# Muted extratropical low cloud seasonal cycle is closely linked to underestimated climate sensitivity in models

Xianan Jiang [1,2] ✉, Hui Su[1,3,4], Jonathan H. Jiang [2], J. David Neelin[4], Longtao Wu[2], Yoko Tsushima [5] & Gregory Elsaesser[6]

A large spread in model estimates of the equilibrium climate sensitivity (ECS), defined as the global mean near-surface air-temperature increase following a doubling of atmospheric $CO_2$ concentration, leaves us greatly disadvantaged in guiding policy-making for climate change adaptation and mitigation. In this study, we show that the projected ECS in the latest generation of climate models is highly related to seasonal variations of extratropical low-cloud fraction (LCF) in historical simulations. Marked reduction of extratropical LCF from winter to summer is found in models with ECS > 4.75 K, in accordance with the significant reduction of extratropical LCF under a warming climate in these models. In contrast, a pronounced seasonal cycle of extratropical LCF, as supported by satellite observations, is largely absent in models with ECS < 3.3 K. The distinct seasonality in extratropical LCF in climate models is ascribed to their different prevailing cloud regimes governing the extratropical LCF variability.

Accurate projection of future climate has been an urgent need to provide scientific guidance for policy-making on climate mitigation and adaptation strategies. Global climate models (GCMs), the primary tools used for climate projection and understanding of climate systems, however, exhibit large uncertainty in depicting how Earth's climate system responds to human activity. For example, the equilibrium climate sensitivity (ECS), a quantity used to represent the change in global mean near-surface air-temperature in response to radiative forcing associated with a doubling of the atmospheric carbon dioxide concentration, significantly varies among GCMs that participated in the World Climate Research Programme (WCRP) Coupled Model Intercomparison Project (CMIP)[1,2]. With increased complexity in climate models, the ECS in GCMs that participated in the latest sixth phase of CMIP (CMIP6) exhibits even a larger spread of 1.8–5.6 °C than that of 2.1–4.7 °C in the CMIP5 models[3–5]. The large discrepancies in ECS among GCMs have been attributed to model uncertainty in

representing various feedback processes, which act to amplify (i.e., a positive feedback) or dampen (a negative feedback) the initial warming. Among them, the short-wave (SW) radiative feedback associated with low clouds, particularly over the extratropical regions of both hemispheres, has been suggested one of the largest sources of uncertainty in predicting the ECS[4–11]. Hereafter, if not stated otherwise, the extra-tropics is referred to the latitude belts of 60–30°S and 30–60°N. All analyses in this study are conducted over ocean grid points to avoid even more complicated feedback processes over the land, and surface temperature is referred to skin temperature if not specifically defined, i.e., sea surface temperature over ocean grid cells. Low cloud fraction (LCF) in both models and observations is derived by vertical cloud fractions below 700 hPa using a maximum overlapping assumption.

As shown in Fig. 1, the strong warming by the end of the 21st century in the high ECS models (ECS greater than 4.75 K,

[1]Joint Institute for Regional Earth System Science and Engineering, University of California, Los Angeles, Los Angeles, CA, USA. [2]Jet Propulsion Laboratory, California Institute of Technology, Pasadena, CA, USA. [3]Department of Civil and Environmental Engineering, Hong Kong University of Science and Technology, Hong Kong, China. [4]Department of Atmospheric and Oceanic Sciences, University of California, Los Angeles, Los Angeles, CA, USA. [5]Met Office Hadley Centre, Exeter, UK. [6]NASA Goddard Institute for Space Studies, and Department of Applied Physics and Mathematics, Columbia University, New York, NY, USA. ✉e-mail: xianan@ucla.edu

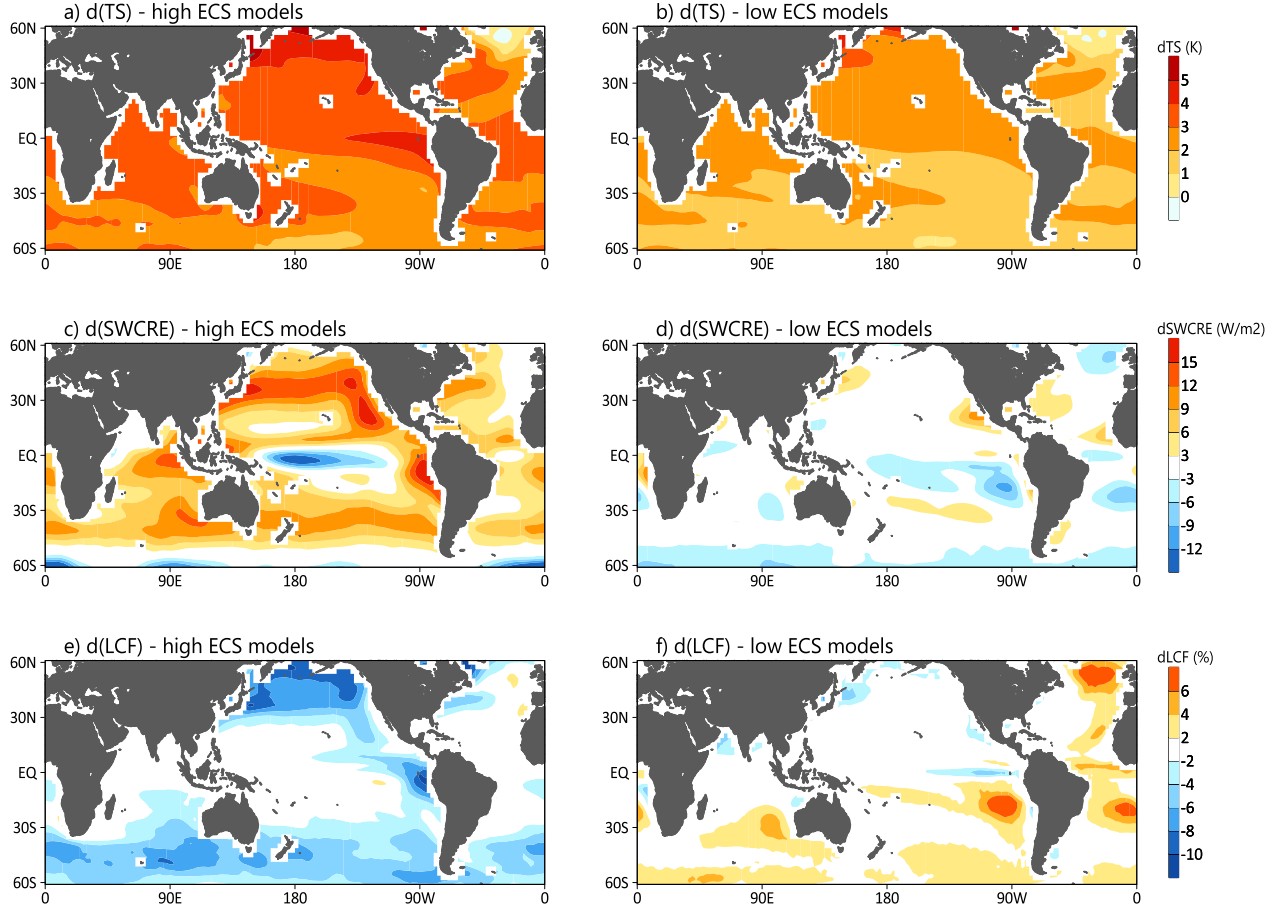

**Fig. 1 | Long-term climate trends in the high and low equilibrium climate sensitivity (ECS) models.** Composite patterns of the differences in climatological annual mean surface temperature (**a**, **b**; Unit: K), top-of-atmosphere (TOA) shortwave (SW) cloud radiative effect (**c**, **d**: SWCRE; Unit: W m$^{-2}$), and low cloud fractions (**e**, **f**; Unit: %) between 2061–2095 from simulations under the shared socio-economic pathway 5-8.5 (SSP585) scenario and 1980–2014 from historical simulations in the high (left columns) and low (right columns) ECS models.

Supplementary Table 1) from CMIP6 (Fig. 1a) is associated with reduced LCF over the extratropical oceans (Fig. 1e), which effectively reduces cloud reflection of solar radiation, indicated by a positive trend in the top-of-atmosphere (TOA) SW cloud radiative effects (SWCRE) (Fig. 1c), and generates excessive heating at the surface. This thus represents a positive cloud feedback that amplifies the surface warming over the extra-tropics. While reduced low clouds with surface warming over tropics or subtropics has been ascribed to a drying effect in the boundary layer due to enhanced vertical moisture gradient and thus mixing[12–14], processes underlying reduced extratropical low clouds under a warming climate remain poorly understood, and are highly relevant to the findings in this study. A slightly equatorward shift in the maximum TOA SWCRE changes relative to the maximum reduction of LCF is noted over 30–60°N/S (c.f. Fig. 1c, e). In addition to weaker solar irradiance near Polar Regions, these equatorward shifts in SWCRE changes can also be due to a negative feedback involved with mixed-phase clouds over the higher latitudes of the extra-tropics[15,16], particularly poleward of 45°N/S. Over these high-latitude extra-tropics, more cloud liquid condensates over ice particles will be formed when the atmosphere warms up, leading to increase of cloud brightness and an enhanced cloud albedo effect, i.e., a surface cooling effect. This negative cloud-phase feedback over the higher latitudes of the extra-tropics can partially offset or even dominate over the positive cloud fraction feedback. A competition between these extratropical low-cloud feedback processes is found to be one of the primary factors responsible for a large spread in ECS simulated in different GCMs[4,8–10,17,18]. The strong positive cloud fraction feedback over the

extra-tropics in both hemispheres as seen in the high ECS models, however, is largely absent in the low ECS models (ECS less than 3.3 K; Fig. 1d, f), consistent with the overall weaker warming trends in these models (Fig. 1b). The low-cloud fraction feedback associated with the long-term climate trend over extratropical oceans, represented by changes in LCF normalized by local sea surface temperature (TS) changes between the 21st century and present-day (see "Methods"), exhibits a high negative correlation (~−0.81) with the model ECS across 18 available GCM simulations (see Fig. 2c, to be further discussed), confirming a critical role of the extratropical low-cloud feedback in contributing to the spread in the projected ECS.

In addition to the extra-tropics, model uncertainties in representing cloud feedbacks over other regions can also be partially responsible for the large inter-model spread in ECS, e.g., over the tropical and subtropical low-cloud regions[8,18–22]. Considering the complexity of physical schemes in the latest climate models and the non-linear interactions among different processes, identification of key model processes governing cloud feedbacks and thus the large spread in model ECS has been greatly challenging. For example, it remains to be determined whether the high or low ECS models are more realistic in their projected warming amplitudes and spatial patterns (e.g., Fig. 1). Addressing this question will be critical to guide the development of strategic plans and policy-making to mitigate impacts of climate change. To assess model fidelity in projecting future climate change, performance of these models in representing various climate processes that are most relevant to climate sensitivity needs to be comprehensively evaluated against available observations such as

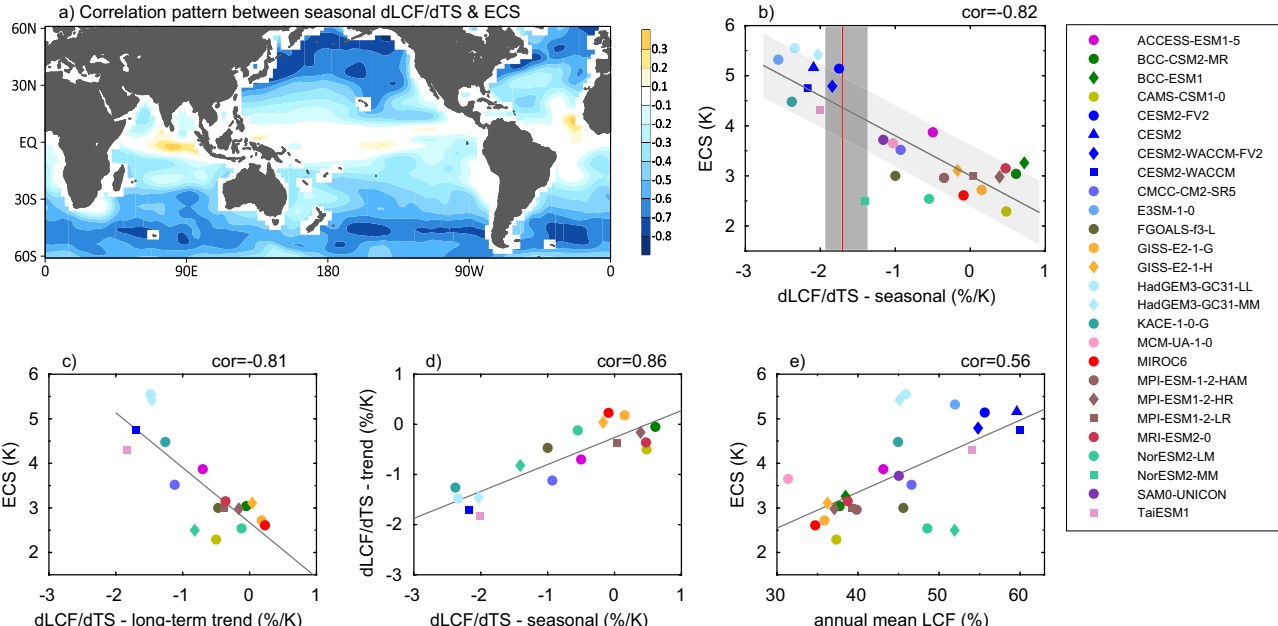

**Fig. 2 | Seasonal cycle of extratropical low-cloud fraction (LCF) as a constraint for model equilibrium climate sensitivity (ECS). a** Spatial pattern of correlations between the ECS and seasonal dLCF/dTS across 26 CMIP6 GCMs based on historical simulations for the period of 1980–2014; Scatter plots of **b** ECS versus the seasonal dLCF/dTS, **c** ECS versus the dLCF/dTS associated with the long-term trend, **d** the seasonal dLCF/dTS versus dLCF/dTS associated with the long-term trend, **e** ECS versus annual mean LCF across multi-model simulations. All variables in the scatter plots (**b**–**e**) except the ECS are derived over the oceanic region between 30–60°N/S. Note that only 18 GCMs are available for the dLCF/dTS associated with the long-term trend in (**c**) and (**d**). The light gray shaded areas around the regression line in

(**b**) represent the standard prediction errors by the linear fit following Schlund et al. (2020)[25]. The vertical red line in (**b**) corresponds to the seasonal dLCF/dTS derived from the climatological seasonal cycle of LCF and TS based on the satellite observations for the period of 2006–2011, with its uncertainty (dark gray shading) estimated by the mean and one standard deviation of the observed seasonal dLCF/dTS in each year of 2006–2011. See "Methods" for details in deriving the dLCF/dTS on various time-scales based on both models and observations. Note that the ECS value predicted by each model used in this study is the "effective climate sensitivity" from Schlund et al.[25], an approximation of the equilibrium climate sensitivity that might be biased[60, 61].

through the process-oriented emergent constraint approach[23–27]. To apply the emergent constraints for climate sensitivity, an empirical relationship between an observable quantity in the past or present-day climate, usually related to some key climate feedback processes, and ECS needs to be identified.

In this study, we propose an emergent constraint on ECS using the seasonal cycle of extratropical LCF based on historical simulations. We show that the extratropical low-cloud fraction feedback associated with the long-term climate trend in CMIP6 model simulations is strongly correlated with the seasonal cycle of extratropical LCF in present-day simulations. A strong seasonal reduction of LCF over the extra-tropics from winter to summer simulated in the high ECS models, which tends to agree better with satellite observations, is in stark contrast to a weak seasonal cycle of LCF in the low ECS models. Possible processes leading to different seasonal variability in LCF between the high and low ECS models are identified.

## Results

### Seasonal cycle of extratropical LCF as a constraint for model ECS

Figure 2a presents a correlation pattern between ECS and the regression slope of seasonal LCF variations against local sea surface temperature (dLCF/dTS) across 26 CMIP6 models (see "Methods"). A strong negative correlation between ECS and the seasonal dLCF/dTS is discerned over a large area of the extra-tropics in both hemispheres. That is, a model with a more rapid decrease in extratropical LCF with increasing local surface temperature on the seasonal time scale tends to produce a higher ECS. When averaging the seasonal dLCF/dTS over the oceanic regions between latitudinal belts of 30–60°N/S in each model, a high correlation of −0.82 between ECS and the seasonal dLCF/dTS is found across the 26 models (Fig. 2b). A strong correlation between ECS and the seasonal dLCF/dTS is also found for northern or

southern hemisphere extratropics only (Supplementary Fig. 1a, b). A rapid decrease of extratropical LCF with surface temperature (<−2%/K) on the seasonal time scale is found in most of the high ECS models, in contrast to a rather weak or even slightly positive dLCF/dTS in the low ECS models. Moreover, the correlation pattern in Fig. 2a greatly resembles the pattern of long-term trend in LCF in high ECS model simulations in Fig. 1e (a pattern correlation -0.6), indicating that processes in regulating seasonal changes of extratropical LCF may also be at play for the low-cloud feedback underlying the future warming. Figure 2d confirms that the dLCF/dTS associated with the long-term trend is indeed highly correlated with the seasonal dLCF/dTS over the extra-tropics, although a more rapid change in LCF with surface temperature is found on the seasonal time scale compared to that associated with the long-term trend. As the dLCF/dTS associated with the long-term trend is closely linked to model ECS (Fig. 2c), this may explain the strong linkage between ECS and the seasonal dLCF/dTS across models in Fig. 2b. Also note that a strong correlation between the seasonal and long-term dLCF/dTS across these models is not dependent on seasons when calculating the long-term dLCF/dTS (Supplementary Fig. 1c, d).

This close relationship between the projected ECS and the seasonal dLCF/dTS over the extra-tropics based on the present-day simulations suggests that the seasonal dLCF/dTS over the extra-tropics can be used as an emergent constraint on ECS. Therefore, the observed seasonal dLCF/dTS over the extra-tropics as derived from satellite observations, e.g., the CloudSat/Cloud-Aerosol Lidar and Infrared Pathfinder Satellite Observations (CALIPSO)[28] and the NOAA Optimum Interpolation Sea Surface Temperature (OISST)[29] (see "Methods"), can provide an important assessment of the confidence in projected ECS in GCMs. In contrast to the rather weak seasonal dLCF/dTS over the extra-tropics in the low ECS models, the observed seasonal dLCF/dTS by

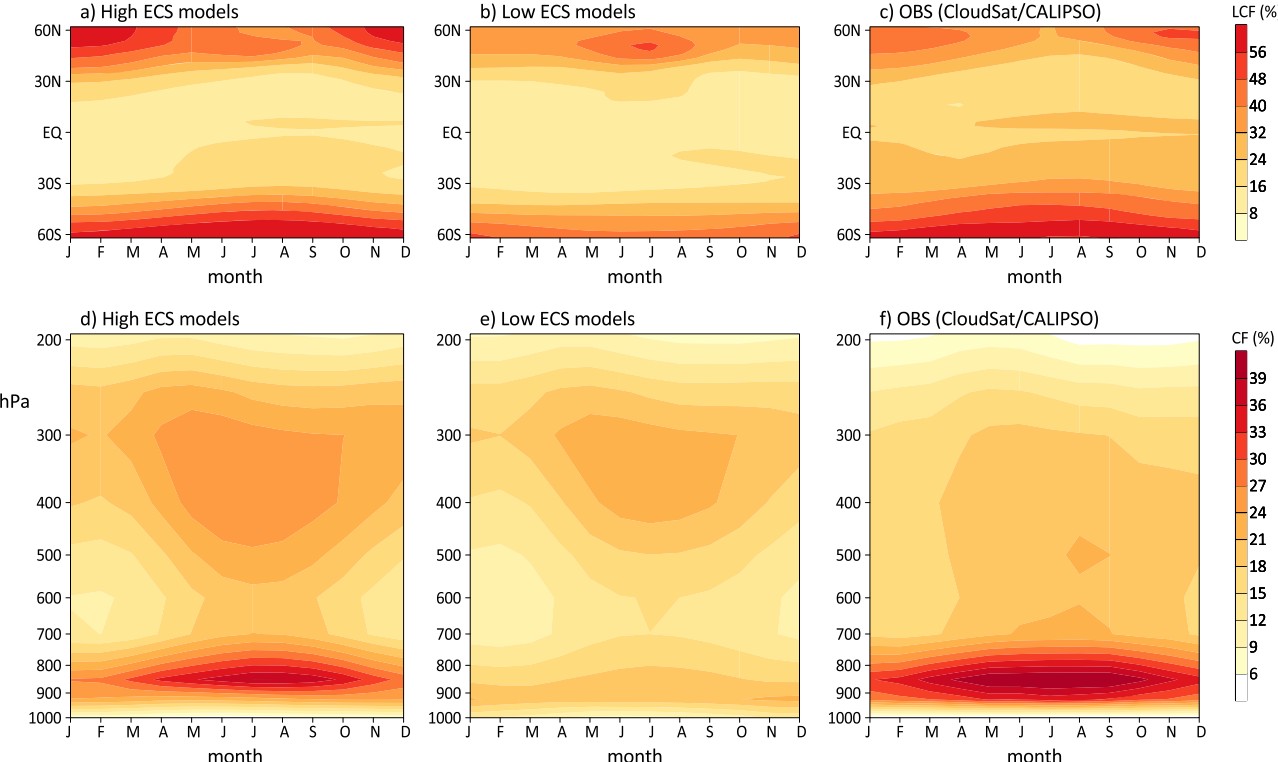

**Fig. 3 | Seasonal variations of extratropical clouds in the high and low equilibrium climate sensitivity (ECS) models. a–c** Composite seasonal cycle of zonal mean low-cloud fraction (LCF) over 60°S–60°N for **a** the high ECS models, **b** low ECS models, and **c** the CloudSat/CALIPSO observations. **d–f** Composite vertical-seasonal cross-sections of zonal mean cloud fractions over the extratropical Southern Ocean (60°S–30°S) for **d** the high ECS models, **e** low ECS models, and **f** CloudSat/CALIPSO observations. Zonal mean values in all these plots are calculated over ocean grid points. All these seasonal evolution patterns are derived based on climatology during 1980–2014 for GCM simulations, and 2006–2011 for the CloudSat/CALIPSO observations (see "Methods").

CloudSat/CALIPSO shows a relatively strong seasonal change of LCF with surface temperature (the vertical red line in Fig. 2b), which is closer to the high ECS models, although the seasonal dLCF/dTS tends to be largely overestimated in several extremely high ECS models. In contrast, the rather weak or even positive seasonal dLCF/dTS as found in the low ECS models is not supported by CloudSat/CALIPSO observations, indicating the projected low ECS values in these models may be underestimated. A moderate ECS has also been suggested by several recent studies based on evaluations of model feedback errors[18,22] or other metrics of emergent constraints[19,30].

Since the seasonal evolution of surface temperature in the high and low ECS models is largely similar (Supplementary Fig. 2b), distinct seasonal dLCF/dTS over the extra-tropics between these two model groups are mainly due to their differences in simulated seasonal variations of extratropical LCF (Fig. 3). A pronounced seasonal cycle in extratropical LCF is discerned in both the high ECS models and CloudSat/CALIPSO observations, with the maximum LCF occurring during the winter season in both hemispheres (Fig. 3a, c), whereas a generally weak seasonal cycle of low clouds is found in the low ECS models (Fig. 3b). In particular, relatively larger low-cloud amounts are found in summer over both hemispheres in the low ECS models, opposite to those in the high ECS models and observations. A pronounced increase of extratropical LCF during the austral winter (May–October) over the Southern Oceans in the high ECS models and satellite observations is further illustrated by seasonal evolution of vertical cloud profiles in Fig. 3d, f with their maximum LCF between 900–800 hPa closely connected to increased clouds in the mid-to-upper troposphere. In the low ECS models, an increase of LCF below 700 Pa during May–October is nearly absent (Fig. 3e).

Due to more LCF during winter in the high ECS models, a larger amount of annual mean LCF is also generally seen in the high ECS

models than those in low ECS models. While the climatological mean LCF over the extra-tropics itself also shows a statistically significant correlation (r ~ 0.56, *p* = 0.0024) with ECS across the models (Fig. 2e), the correlation is relatively weaker compared to that between the seasonal dLCF/dTS and ECS (Fig. 2b). We therefore consider the seasonal dLCF/dTS over the extra-tropics a better metric to constrain the model ECS than the climatological annual mean LCF.

## Short-term trend in extratropical low-cloud fractions

In addition to the long-term trend and seasonal cycle, Fig. 4 further demonstrates that the distinct extratropical low-cloud feedbacks between the high and low ECS models can also be clearly detected in the trend of LCF during a relatively short period of 1980–2014 in historical simulations. Despite their largely similar surface temperature trend (Fig. 4b), a significant reduction of extratropical LCF during 1980–2014 is clearly seen in the high ECS models, but not in the low ECS models (Fig. 4a). Differences in the trend of the extratropical LCF between these two model groups are largely consistent with the trend in TOA SWCRE (Fig. 4d). While an increasing trend in SWCRE during 1980–2014 is found in the high ECS models, which tends to be supported by the Clouds and the Earth's Radiant Energy System (CERES) satellite observations[31], almost no trend is detected in the low ECS models (Fig. 4d).

The low-cloud feedback (dLCF/dTS) derived by the linear regression slope of the annual mean LCF and TS over the extra-tropics during 1980–2014 is found to be highly correlated to the dLCF/dTS associated with the long-term trend (Fig. 4e), to the seasonal dLCF/dTS (Fig. 4f), and also to the ECS (Fig. 4g). A significant negative correlation between the dLCF/dTS associated with the short-term trend and ECS (r = −0.69, *p* < 0.001) is still evident, although slightly weaker than the correlations with the dLCF/dTS based on the seasonal cycle (Fig. 2b)

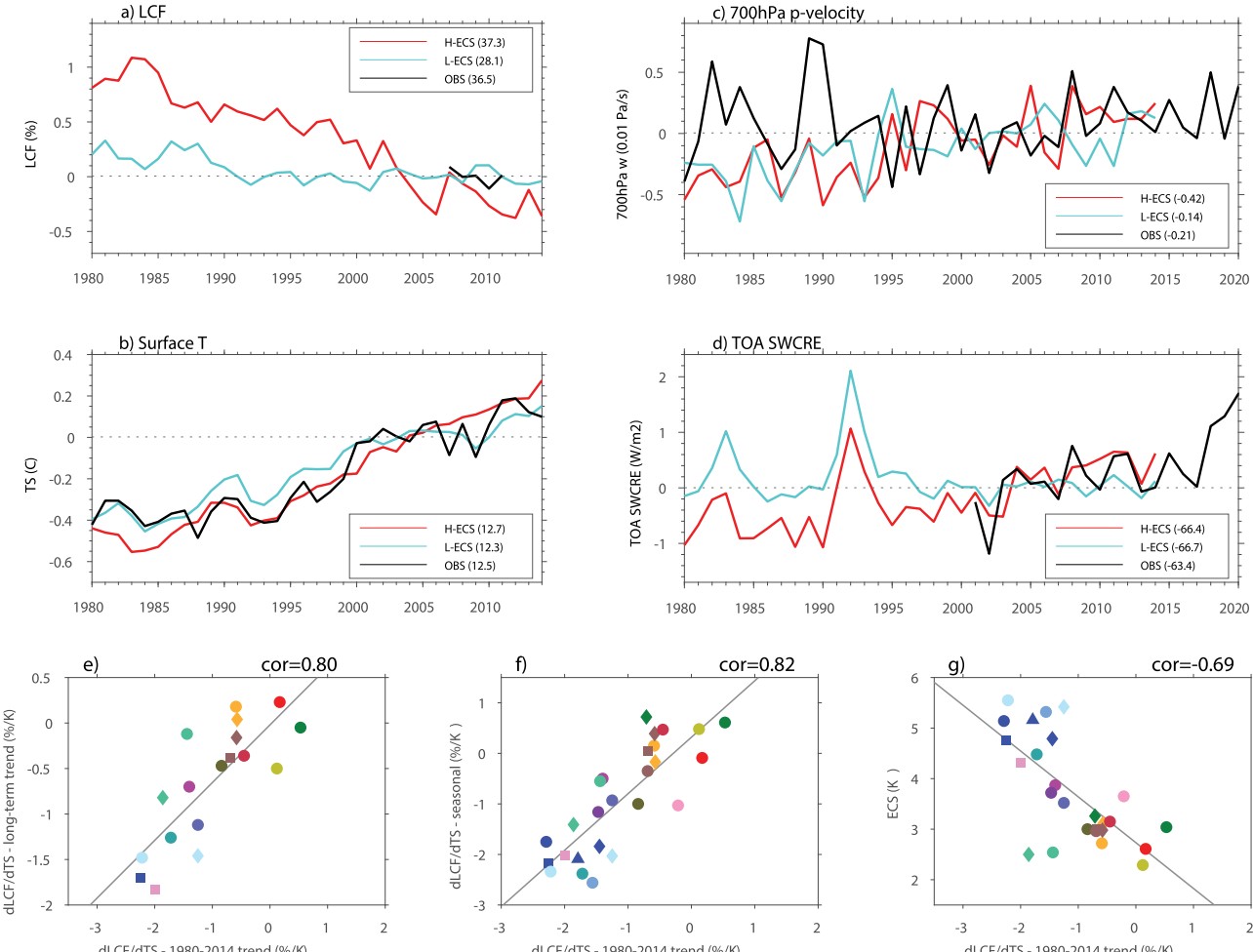

**Fig. 4 | Short-term climate trends over the extratropics in the high and low equilibrium climate sensitivity (ECS) models.** Evolution of low-cloud fraction (LCF) (**a**; Unit: %), surface temperature (**b**; Unit: °C), p-vertical velocity at 700 hPa (**c**; Unit: 0.01 Pa S⁻¹), and the top-of-atmosphere (TOA) short-wave (SW) cloud radiative effect (SWCRE) (**d**; Unit: W² m⁻¹) over the extra-tropics during 1980–2014 for the high (red) and low (cyan) ECS model composites, and the observations (black; observations for LCF are based on CloudSat/CALIPSO, surface temperature based on NOAA OISST, vertical p-velocity based on the ERA5 reanalysis, and TOA SWCRE based on CERES). Note that in (**a**–**d**), the corresponding climatological values for each value, denoted by the numbers in the parentheses following the high or low ECS models and observations in the legend of each panel, are removed from the time series. **e**–**g** Scatter plots between the dLCF/dTS based on the short-term trend during 1980–2014 and **e** the dLCF/dTS based on the long-term trend, **f** the seasonal dLCF/dTS, and **g** ECS in available CMIP6 model simulations. All variables in this figure except ECS are derived over the extratropical ocean grids between 30–60° in both hemispheres. See the legend in Fig. 2 for the model name corresponding to each mark shown in (**e**–**g**).

and long-term trend (Fig. 2c), which can be partially ascribed to the influences of the internal climate variability considering a relatively short period of 1980–2014, such as the decadal or inter-decadal variability as the surface warming patterns also matter for cloud feedbacks[32–34]. These results lend further credence that the significant discrepancies in the extratropical low-cloud feedback between the high and low ECS models can be robustly identified based on the seasonal cycle and short-term trend of extratropical LCF from historical simulations. In particular, considering that a climatological seasonal cycle can be readily derived in the present-day simulations, the seasonal dLCF/dTS can thus provide an especially useful emergent constraint on ECS. Moreover, understanding of the underlying processes responsible for the distinct seasonal LCF variability between the high and low ECS models can provide important insights to their differences in predicted ECS.

**Cloud regimes associated with the extratropical LCF variability**
Despite the distinct seasonal variations in extratropical LCF between the high and low ECS models, several large-scale cloud controlling factors that have been widely used to understand low-cloud

variations[12,14], including surface temperature, the estimated inversion strength (EIS), lower-tropospheric relative humidity and vertical velocity, exhibit largely similar seasonal evolution features between the two model groups (Supplementary Fig. 2). These large-scale factors thus do not readily explain the key differences in their seasonal LCF variations. Since extratropical clouds are strongly linked to transient extratropical cyclone activity on synoptic time-scales[35–39], model deficiencies in representing clouds embedded in extratropical cyclones can lead to significant biases in the simulated cloud-radiation feedback and thus climate sensitivity[40–42]. A cloud regime-based approach is thus applied to understand the different characteristics in extratropical low clouds between the high and low ECS models. As daily vertical cloud profiles are needed to derive prevailing synoptic cloud regimes, which are only available from very limited models that participated in CMIP6, the cloud regime-based analyses in this section are based on three GCMs, with two high ECS models, CESM2 and HadGEM3-GC31-LL, and one low ECS model, MPI-ESM1-2-LR, as representatives for the high and low ECS models, respectively. These three models exhibit very typical characteristics of the seasonal cycle as well as the climate trend of

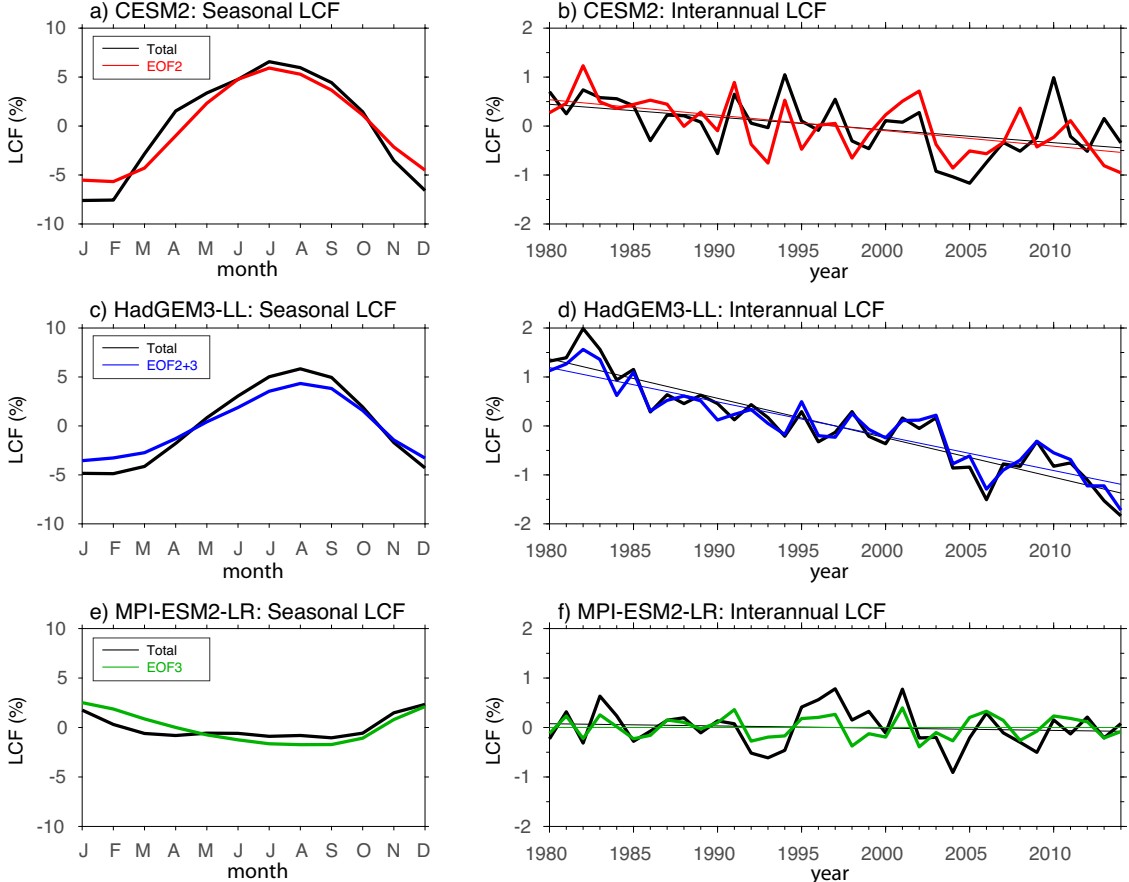

**Fig. 5 | Leading cloud regimes associated with the extratropical low-cloud fraction (LCF) variability. a**, **b** Climatological seasonal cycle of LCF (**a**) and the time series of annual mean LCF (**b**) over the southern hemisphere extra-tropics (45–55°S), shown as deviations from their long-term climatology during the period of 1980–2014, in CESM2 simulations (black) along with reconstructions using the daily LCF variability associated with the mid-top cloud regime (i.e., the Empirical Orthogonal Function, EOF, mode #2, red). **c**, **d** Same as in the upper panel but the black lines for HadGEM3-LL simulations and blue lines for reconstruction using the daily LCF variability associated with the combined mid- and low-top cloud regimes (EOF modes #2 and #3). **e**, **f** Same as in the upper panels but the black lines for MPI-ESM2-LR simulations and green lines for reconstruction using the daily LCF variability associated with the low-top cloud regime (EOF mode #3). See "Methods" for details on the derivation of the leading extratropical cloud regimes based on the EOF analysis.

extratropical low clouds in the high ECS and low ECS models (Supplementary Fig. 3, also Fig. 5 to be discussed).

With a focus over the Southern Oceans, prevailing extratropical cloud regimes in the three GCMs are derived based on an empirical orthogonal function (EOF) analysis of daily vertical profiles of cloud fractions (see "Methods"). Despite slight differences in their detailed structures, the three leading modes of daily vertical cloud variability derived from the three models are largely similar, representing the high-, mid-, and low-top cloud regimes, respectively (Supplementary Fig. 4). The mid-top cloud mode, which is strongly coupled with the lower-to-mid-tropospheric ascending motion, bears a great resemblance to the cloud structure associated with the observed extra-tropical cyclones[36,38,39], characterized by a gradual transition from low clouds over the cold front sector (west side) to deep clouds over the warm front sector (east side) of cyclones. This suggests a strong modulation of low clouds by extratropical cyclones along the mid-latitude storm tracks[38,40,41,43]. Since the seasonal cycle and interannual extratropical LCF during 1980–2014 in these three models can be well reproduced using their corresponding long-term climatology plus variability associated with the three leading modes (Supplementary Fig. 5, Fig. 5), this makes it possible to identify the dominant cloud regime controlling the variability of extratropical low clouds in these models. Significantly enhanced LCF over extratropical Southern Oceans during the austral winter (May–October) is found to be mainly contributed by the mid-top cloud regime in CESM2 (Fig. 5a), and by the

combined mid- and low-top cloud regimes that are in-phase with each other in HadGEM3-LL (Fig. 5c, and Supplementary Fig. 5i, j). This suggests that enhanced tropospheric vertical velocity associated with the active storm-track variability, fueled by the baroclinic instability that is most energetic during the winter season[44,45], may play a crucial role in generating the maximum extratropical LCF during the austral winter in CESM2 and HadGEM3-LL. In contrast, while LCF associated with the mid-top and high-top cloud regimes also exhibits a maximum during the austral winter, they only play minor roles in regulating seasonal variations of low clouds in MPI-ESM2-LR (Supplementary Fig. 5); instead, its seasonal cycle is dominated by the low-top cloud regime, which favors a maximum in the austral summer (November–March) and a minimum in winter (May–October; Fig. 5e), largely in concert with seasonal evolution of the lower-tropospheric instability, e.g., the EIS (Supplementary Fig. 2).

The distinct vertical cloud variability in the three models is readily seen by a snapshot of seasonal evolution of vertical cloud profiles over a representative region of southern extratropics during a randomly selected year (Supplementary Fig. 6). Significantly enhanced LCF or more frequent occurrence of low clouds during the austral winter (May–October) is found to be closely linked to vertically-extended clouds in the mid- and upper-troposphere in CESM2 and HadGEM3-LL, as also largely evident in CloudSat/CALIPSO observations. A crucial role of the mid-top cloud regime associated with extratropical cyclones for the radiative budget over the extra-tropics has also been

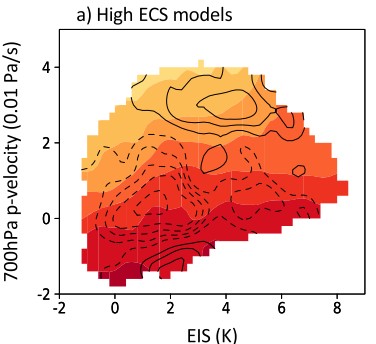
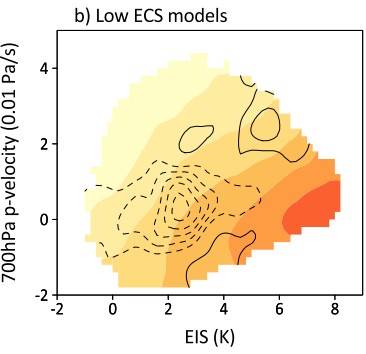
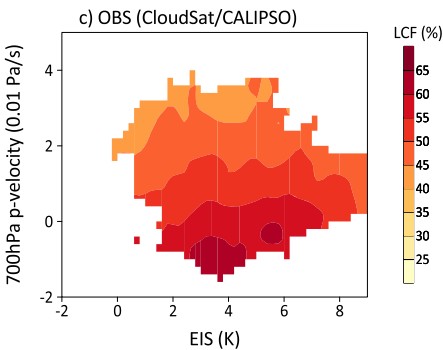

**Fig. 6 | Cloud controlling factors for extratropical low-cloud fraction (LCF) in the high and low equilibrium climate sensitivity (ECS) models.** Composite LCF (shading; unit: %) as a function of vertical p-velocity at 700 hPa (Unit: 0.01 Pa s$^{-1}$) and the estimated inversion strength (EIS; unit: K) based on **a** the high ECS models, **b** low ECS models, and **c** CloudSat/CALIPSO observations. These composites are derived based on monthly mean fields of climatological seasonal cycle at all ocean grid points between 30–60° over both hemispheres. Model composite are based on historical simulations from 1980–2014, while observations are based on the period of 2006–2011. Contours in (**a**) and (**b**): differences in the frequency of occurrence of LCF in model simulations between future climate (21$^{st}$ century simulations for the period of 2060–2094 under the SSP585 scenario) and historical simulations from 1980–2014 (solid/dashed lines for increased/reduced frequency of occurrence under the future climate with the first contour for ±0.1% and an interval of 0.1%).

previously reported in both observations and CMIP5 model simulations[10]. In contrast, slightly reduced low clouds particularly near the surface during May–October are found in MPI-ESM2-LR despite the enhanced deep clouds in general associated with extratropical cyclone activity, confirming a decoupling of LCF from the vertically extended clouds.

Similar as the seasonal cycle, the mid-top or the combined mid- and low-top cloud regimes are also found to play a dominant role in determining the decreasing trend in extratropical LCF during the period of 1980–2014 in CESM2 and HadGEM3-LL (Fig. 5b, d). Considering a strong coupling between the lower-to-mid-tropospheric ascending motion and the vertically-extended clouds (Supplementary Fig. 4), a significant decreasing trend in the extratropical LCF during 1980–2014 in CESM2 and HadGEM3-LL and other high ECS models (Fig. 4a) could be closely linked to the decreasing trend in the mid-tropospheric ascending motion as illustrated in Fig. 4c. While a decreasing trend in the extratropical ascending motion with recent warming is also evident in the low ECS models including MPI-ESM2-LR (Fig. 4c), the change in tropospheric vertical motion tends to not effectively affect low clouds in these models due to a decoupling between LCF and the vertically-extended clouds as previously discussed. As a result, the year-to-year variability of extratropical low clouds in the low ECS models such as MPI-ESM2-LR is largely controlled by the low-top cloud regime as in its seasonal cycle, leading to the absence of a significant decreasing trend in extratropical LCF during 1980–2014 (Figs. 4a, 5f). Note that while a decreasing trend in the ascending motion over the extratropics is simulated in both the high and low ECS models, it is not evident in the ERA5 reanalysis (Fig. 4c), possibly due to a lack of observational constraints involved with the vertical velocity field in the reanalysis dataset.

These above results suggest that the distinct extratropical low-cloud feedbacks that are closely linked to their projected ECS between the high and low ECS models, could be largely due to their differences in depicting the coupling between tropospheric circulation and extratropical LCF. A pronounced reduction of LCF with decreasing tropospheric ascending motion is found in the high ECS models through a strong modulation by the vertically-extended clouds, whereas there is a lack of such strong coupling between tropospheric circulation and low clouds in the low ECS models. Instead, in the low ECS models the variability of extratropical low clouds is dominated by the low-top cloud regime that largely follows the lower-tropospheric stability as suggested by its seasonal cycle.

Motivated by the above cloud regime analyses, following a similar approach used in Grise and Medeiros (2016)[46], distinct processes involved with the LCF variability between the high and low ECS models are further elaborated by composite LCF as a function of vertical velocity at 700 hPa and the EIS over each ocean grid point of the extra-tropics in both hemispheres based on their climatological seasonal cycles (Fig. 6, shaded contours). A strong dependence of LCF on 700 hPa vertical velocity is clearly evident in the high ECS models and CloudSat/CALIPSO observations (Fig. 6a, c), with more (less) low clouds over regions with strong ascending (descending) motions. In contrast, LCF in the low ECS models exhibits a strong dependence on EIS with more (less) LCF associated with high (low) EIS, particularly when ascending motion prevails in the lower-troposphere (Fig. 6b, shaded). During a seasonal transition from winter to summer, changes in LCF due to the increase in the EIS while decrease in the ascending motion tend to offset each other, leading to a rather weak seasonal cycle of LCF in the low ECS models. Projection of the occurrence frequency of LCF in the 21st century simulations suggests that the reduced LCF in the high ECS models is associated with more frequent occurrence of strong descending motion, while no significant changes in LCF in the low ECS models are due to a cancellation with enhanced EIS and increased descending motion under the future climate (Fig. 6, contours), in accord with the climate trend in LCF simulated in the high and low ECS models (Fig. 4a).

## Discussion
Despite the urgent need for accurate climate projection to guide the development of climate mitigation and adaptation strategies, our state-of-the-art climate models exhibit large uncertainties in predicting the magnitude of future warming, as measured by the ECS. Understanding the underlying processes responsible for the large inter-model spread in ECS and constraining ECS with available observations are thus critical for model improvement. Motivated by recent studies on the crucial role of the extratropical low-cloud radiative feedback for climate sensitivity, in this study we propose a metric based on seasonal variations of extratropical LCF to constrain model ECS. We show that seasonal changes of LCF with surface temperature, dLCF/dTS, over the extratropical oceans between 30–60° in both hemispheres, is closely related to the low-cloud feedback under long-term climate change, and is thus highly correlated to model ECS. A strong negative value of seasonal dLCF/dTS over the extra-tropics, i.e., a significant reduction of LCF from winter to summer indicative of a strong positive cloud feedback, is found in the high ECS models, in contrast to rather weak seasonal variations of LCF in the low ECS models. Strong seasonal reduction of LCF over the extra-tropics from winter to summer in the high ECS models is in general agreement with the CloudSat/CALIPSO

observations, suggesting that the predicted low ECS values in climate models may be underestimated.

Based on three GCMs, CESM2, HadGEM3-LL, and MPI-ESM2-LR, as representatives of the high and low ECS models, a cloud regime analysis is performed to identify key processes underlying distinct extratropical cloud feedbacks in these models. In the high ECS models, the LCF variability tends to be dominated by a mid-top or a combined mid- and low-top cloud regime that is strongly coupled to the lower-to-mid-tropospheric vertical velocity. During the seasonal migration from winter to summer, tropospheric ascending motion over the extra-tropics weakens due to reduced baroclinicity and storm-track variability, and leads to a significant reduction of extratropical LCF in these models. Under a warming climate, weakening of the ascending motion over the extra-tropics, possibly associated with the expansion of the Hadley Cell[47–50] and/or the poleward shift of mid-latitude jet stream[45,51], can also lead to reduced extratropical LCF in the high ECS models through modulation of the vertically-extended clouds, therefore, a positive low-cloud feedback that leads to rapid warming over the extratropics in these models. In contrast, in the low ECS models as indicated by MPI-ESM2-LR, the variability of extratropical low clouds is mainly dominated by a low-top cloud regime, which exhibits a seasonal cycle that largely follows the lower-tropospheric stability, i.e., with a maximum in summer and a minimum in winter, and lacks a significant trend under a warming climate.

Since the emergent relationship between the seasonal dLCF/dTS over the extratropics and ECS in this study was derived based on CMIP6 models, it is interesting to verify whether this metric is also applicable for CMIP5 models. By plotting the seasonal dLCF/dTS and their corresponding ECS from 21 CMIP5 models (Supplementary Table 2) along with CMIP6 models (Supplementary Fig. 7), it is found that a majority of the CMIP5 models exhibit a much weaker seasonal cycle of extratropical LCF than the observations and the high ECS models from CMIP6, consistent with generally low ECS in the CMIP5 models[4]. While a statistically significant correlation between ECS and the seasonal dLCF/dTS over the extratropics can still be obtained based on 26 CMIP6 and 21 CMIP5 models (r ~ −0.56, $p < 0.0001$), no significant correlation is found across the CMIP5 models alone (r ~ 0.06). This result suggests that the high ECS predicted in several CMIP6 models, which largely leads to the increase in the spread of ECS from CMIP5 to CMIP6, is closely associated with changes in representation of extratropical low-cloud feedbacks, consistent with findings from other recent studies[4,8,10].

The cloud regime analysis suggests that the rapid reduction of extratropical low clouds under a warming climate in the high ECS models could be mainly due to the reduction of vertically-extended clouds in association with the weakening of ascending motion over the extratropics. However, future investigations are needed to better understand how the reduced vertical clouds and associated weakening of the ascending motion over the extratropics under a warming climate are linked to changes of the Hadley Cell expansion, the mid-latitude storm-track variability, and the shift of the westerly Jet Stream. A weak relationship between extratropical SWCRE or ECS and poleward shifts of the extratropical jet streams has been previously reported in observations and CMIP5 models[45,52]. On one hand, this could be due to a significant role of the Hadley Cell expansion on extratropical LCF in addition to the shift of the Jet Stream[52]. Meanwhile, this can also be ascribed to different model responses of extratropical low clouds to changes in environmental conditions as suggested by this study. For example, a weakening of the extratropical ascending motion is simulated in both the high and low ECS models (Fig. 4c), whereas the reduced extratropical LCF is only simulated in the high ECS models (Fig. 4a). Since CMIP5 models exhibit large deficiencies in representing the seasonal evolution of extratropical low clouds in general, investigations based on the high ECS models from CMIP6 are

expected to provide important insights on how extratropical LCF and its induced radiative effects respond to changes of the dynamical processes over the extratropics. Also note that considering large model deficiencies in representing extratropical storm-track clouds in the low ECS models as shown in this study, it needs to be cautious when assessing environmental changes over the extratropics under a future climate based on these low ECS models, including most of the CMIP5 models.

In this study, we mainly focus on the low-cloud fraction feedback over the extra-tropics. It has been suggested that model uncertainties in representing the cloud-phase feedback can also be related to large model spread in predicted global climate sensitivity as previously discussed[4,8–10,53], although with a possible secondary role compared to the cloud-fraction feedback over the extra-tropics between 30–60°N/S[4]. An examination of the liquid condensate fraction as a function of atmospheric temperature, an important metric indicating the fraction of super-cooled liquid water in mixed-phase clouds[4,16], does not show a significant difference between the high and low ECS models (Supplementary Fig. 8). A detailed investigation of SW radiative feedback over the extra-tropics due to the cloud phase and cloud optical-depth feedbacks between the high and low ECS models is warranted in future studies. We also note that a model's overall climate sensitivity can be determined by multiple feedback processes over the globe in addition to the extratropical low-cloud feedback examined in this study[18,22,25]. For example, several other cloud-based emergent metrics have been recently proposed to constrain model climate sensitivity, including tropical shallow cumulus and stratocumulus clouds[19], seasonal cycle of subtropical marine LCF[21], or the regime-averaged marine low-cloud feedbacks between 60°S–60°N[20]. The emergent relationship between model ECS and the seasonal variability of extratropical LCF proposed in this study provides a unique metric to quantify model representation of low-cloud feedbacks over the extratropics.

## Methods

### The dLCF/dTS at various time-scales in model simulations

Model low cloud fraction (LCF) on each grid cell is derived by the vertical profile of cloud fractions below 700 hPa using a maximum overlapping assumption. The climatological seasonal evolution of LCF from January to December is obtained from monthly mean LCF for the period of 1980–2014 based on historical simulations from CMIP6. The slope of LCF variations as a function of surface temperature (dLCF/dTS) on the seasonal time scale at each grid point is calculated based on the 12 monthly values of LCF and TS from their climatological seasonal cycle using a linear regression following Zhai et al.[21]. Seasonal dLCF/dTS over the extratropics in each model as shown in Fig. 2b is obtained by averaging the dLCF/dTS over all ocean grid points between 30–60°N/S.

Similarly as for the seasonal dLCF/dTS, the dLCF/dTS associated with the short-term trend during the period of 1980–2014 on each grid cell is derived by a linear regression of annual mean LCF and surface temperature during the period of 1980–2014. Then the dLCF/dTS associated with the short-term trend over the extra-tropics in each model as shown in Fig. 4e, f is obtained by averaging dLCF/dTS over all ocean grid points between 30–60°N/S.

To calculate the dLCF/dTS associated with the long-term trend over the extra-tropics in each model as shown in Fig. 2c, d, long-term changes in LCF (dLCF) and surface temperature (dTS) on each grid cell are first derived by the differences in their 35-year mean values between 2061–2095 from simulations under the shared socio-economic pathway 5-8.5 (SSP585) scenario and 1980–2014 from historical simulations. Then dLCF and dTS are further averaged over all ocean grid points between 30–60°N/S, and the ratio of their spatially averaged values is defined as the dLCF/dTS associated with the long-term trend.

## Observational datasets

The latest ERA5 reanalysis from the European Centre for Medium-Range Weather Forecasts (ECMWF)[54] is used to characterize large-scale cloud controlling factors associated with variability of extratropical low clouds. The observed monthly mean surface temperature used for this study is based on the NOAA Optimum Interpolation Sea Surface Temperature (OISST) dataset (version 2)[29]. Observed TOA all-sky and clear-sky fluxes from the Clouds and the Earth's Radiant Energy System (CERES) instrument (version EBAF Ed4.1)[31] is also used to verify model simulations.

The observational dataset for vertical cloud profiles is based on a combined product from the CloudSat and Cloud-Aerosol Lidar and Infrared Pathfinder Satellite Observation (CALIPSO) satellites (2B-GEOPROFLIDAR; version P2R05)[28]. The combined CloudSat/CALIPSO dataset has been considered the best satellite observations for vertical cloud structures associated with extratropical cyclones[36,43,55,56], mainly due to its advantage of the Cloud Profiling Radar (CPR) aboard CloudSat in detecting optically thick hydrometeor layers, and the CALIPSO lidar in detecting optically thin cloud layers that could be missed by the CPR[28,57,58].

Due to a severe anomaly of CloudSat occurred in 2011, Cloud-Sat/CALIPSO vertical cloud profiles during the period of 2006–2011 are used in this study. Similarly as for model results, LCF on each grid cell is defined by the vertical cloud fractions below 700 hPa using a maximum overlapping assumption. While a seasonal dLCF/dTS of −1.71%/K over the extratropics from the CloudSat/CALIPSO observations is derived based on climatological seasonal cycle of vertical clouds (Fig. 2b, vertical red line), its corresponding value derived from each individual year during the 6-year period shows relatively weak year-to-year variability with a mean value of −1.65%/K and a standard deviation of 0.27%/K. This mean value of the seasonal dLCF/dTS along with the one standard deviation value is used to estimate uncertainties involved with the observed seasonal dLCF/dTS (see the vertical gray bar in Fig. 2b). Sensitivity tests also suggest that the climatological seasonal cycle of extratropical LCF and thus the seasonal dLCF/dTS derived based on model simulations during the 6-year period of 2006–2011 are largely identical to that derived from the entire 35-year period of historical simulations as shown in Fig. 3. This lends confidence in constraining the simulated seasonal dLCF/dTS over the extratropics using the CloudSat/CALIPSO observations.

It is noteworthy that global low cloud observations are also provided by several passive-sensing satellites, such as the International Satellite Cloud Climatology Project[59]. While these passive instruments provide useful information of horizontal distribution of low-top clouds, they have limitations in detecting low clouds collocated with deep or multi-layered clouds, such as those associated with extratropical cyclones along the mid-latitude storm tracks[28,36,43,56]. The passive sensing also has difficulties in accurately detecting the cloud-top height[40]. Therefore, the low-cloud fractions based on these passive-sensing satellite observations are not supposedly to be directly compared to those derived from the active CloudSat/CALIPSO observations and model simulations in this study.

## Cloud regime analysis

An Empirical Orthogonal Function (EOF) analysis is used to objectively identify the leading cloud regimes over the extra-tropics with a particular focus over the Southern Oceans. Before the EOF analysis, we divide the global longitude belts between 45–55°S into 12 sub-regions with equal areas. Namely, each sub-region covers an area of 30 × 10 longitude-latitude degrees, representing a typical size of an extratropical cyclone. Daily vertical profiles of cloud fractions on 19 pressure levels between 1000 hPa and 1 hPa spatially averaged over each sub-region are then obtained for the period of 1980–2014 (total 12,775 days). An EOF analysis is then conducted based on the covariance matrix of a concatenated daily series of vertical cloud fraction anomalies over the 12 sub-regions (after removal of their corresponding long-term climatology over each sub-region) during the 35-year period, i.e., with total 19 spatial points (i.e., vertical levels) and 153,300 temporal points (12,775 days × 12 sub-regions, the first 12,775 time points for the 1st sub-region, the second 12,775 time points for the 2nd sub-region, and so on). The derived eigenvectors based on the EOF analysis depict the vertical cloud profiles associated with the leading extratropical synoptic-scale variability modes, and the corresponding principal components (PCs) similarly contain a concatenated daily time series during the 35-year period for each sub-region. Evolution of vertical cloud variability over each sub-region associated with each leading mode or multiple mode combinations can then be reconstructed based on the eigenvectors and their corresponding daily PC coefficients along with their long-term climatological vertical cloud profile. Daily zonal mean vertical cloud profiles over the entire extra-tropics between 45–55°S contributed by each leading mode or multi-mode combinations can be further obtained by averaging the reconstructed daily profiles over the 12 sub-regions. Moreover, detailed 3D structures in cloud and vertical velocity anomalies associated with each leading mode can also be derived based on lag-0 regressions of these daily anomalous fields (2.5 × 2.5 deg, 19 pressure levels, similarly with their corresponding long-term climatology on each grid removed) against the corresponding daily PCs over each sub-region, and then by compositing these structures over the 12 sub-regions with respective to the center of each sub-region, e.g., longitude 0 in Supplementary Fig. 4. In this study, EOF analyses of anomalous vertical clouds over the South Oceans are conducted independently for CESM2, Had-GEM3-LL, and MPI-ESM2-LR, with their corresponding eigenvectors of the first three leading modes shown in Supplementary Fig. 4a, e, i, respectively. Note that a similar EOF analysis can also be conducted by including extratropical sub-regions over the north Pacific and north Atlantic between 45–55°N, the leading modes of synoptic-scale cloud variability are largely unchanged (not shown). The leading EOF modes are also largely similar by slightly modifying the latitudinal or longitudinal width of the extratropical sub-regions.

## Data availability

The CMIP6 GCM outputs were downloaded from the ESGF https://esgf-node.llnl.gov/search/cmip6. The ERA5 data was downloaded from the website https://cds.climate.copernicus.eu/cdsapp#!/dataset/reanalysis-era5-pressure-levels. The CloudSat/CALIPSO cloud data was downloaded from https://www.cloudsat.cira.colostate.edu/data-products/2b-geoprof-lidar. CERES-EBAF TOA fluxes were downloaded from https://asdc.larc.nasa.gov/project/CERES/CERES_EBAF_Edition4.1. NOAA OISSTv2 data was downloaded from https://psl.noaa.gov/data/gridded/data.noaa.oisst.v2.html. The processed climatological monthly mean LCF and TS from historical simulations of CMIP6 models and satellite observations used for derivation of the seasonal dLCF/dTS can be downloaded from https://ucla.box.com/v/ecs-vs-extratropical-lcf.

## Code availability

The Fortran codes to derive the correlation pattern between seasonal dLCF/dTS and model ECS (Fig. 2a), and the correlation between seasonal dLCF/dTS over the extratropics and model ECS (Fig. 2b) can be downloaded from https://ucla.box.com/v/ecs-vs-extratropical-lcf. Other codes used to produce the figures and to derive dLCF/dTS associated with climate trend and the leading extratropical cloud regimes are available from the corresponding author upon request.

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

## Acknowledgements

We acknowledge support by the NSF Climate Process Team Program through grant AGS-1916619 (X.J.), NOAA MAPP Program through grant NA20OAR4310394 (H.S., J.D.N.) and DOE RGMA Program through grant DE-SC0021312 (H.S., J.D.N.), the NASA Obs4MIPS Project through Task Order 80NM0018D0004 (J.H.J.), and the Jet Propulsion Laboratory, California Institute of Technology, under contract with NASA. X.J. acknowledges J. Teixeira for stimulating discussions during this study, and M. Lebsock, W. Bertrand, J. Kay for helpful discussions on the CloudSat/CALIPSO data.

## Author contributions

X.J. and H.S. conceived the study. X.J. conducted the analysis, and led the writing of this manuscript. H.S., J.H.J., J.D.N., L.W., Y.T., and G.E. contributed to discussions during this study and edited the manuscript.

## Competing interests

The authors declare no competing interests.
