## [Peer Review File · Nature Communications]

Muted extratropical low cloud seasonal cycle is closely linked to underestimated climate sensitivity in modelsREVIEWER COMMENTS

Reviewer #1 (Remarks to the Author):

Review on "Absence of extratropical low cloud seasonal cycle is closely linked to an underestimate of climate sensitivity in computer models"

The paper "Absence of extratropical low cloud seasonal cycle is closely linked to an underestimate of climate sensitivity in computer models" by Xianan Jiang et al. provides an emergent constraint on equilibrium climate sensitivity (ECS) based on the seasonal variation of the extratropical low cloud fraction (LCF). They find that models with a strong reduction of the extratropical LCF from winter to summer also show a strong reduction of the extratropical LCF under global warming (thus, a positive low cloud feedback) and therefore have a high ECS. In contrast, models without that pronounced seasonal cycle in extratropical LCF generally have a low ECS. Since this strong seasonal variation of extratropical LCF found in the high-ECS models is also present in satellite observations, the authors conclude that ECS is currently underestimated in modern climate models and that the high-ECS models are more consistent with observations.

In addition, the authors provide further evidence of this connection between the seasonal and long-term LCF variation using an EOF-based analysis of cloud regimes associated with low cloud (LC) variability in the high- and low-ECS models. In the high-ECS models and the satellite observations, the LC variability is dominated by a mid-level cloud regime coupled to lower-to-mid tropospheric vertical motion and associated with extratropical cyclones. From winter to summer, extratropical tropospheric ascending weakens, which leads to a reduction of extratropical LCF. Similarly, a weakening of the ascending motion with global warming through an expansion of the Hadley cell and a poleward shift of mid-latitude jet stream also leads to a reduction of the LCF, forming this strong positive low cloud feedback in the high-ECS models. On the other hand, for low-ECS models, the LC variability is dominated by a low-level cloud regime whose seasonal cycle follows the lower-tropospheric stability (maximum in summer; minimum in winter) and thus is missing a strong trend with global warming. This provides further evidence for an underestimation of ECS in modern climate models.

Reducing uncertainties in future warming and/or climate sensitivity is crucial to define optimal climate mitigation and adaptation strategies. I agree with the authors that emergent constraints based on strong and robust physical mechanisms like the one advertised in this study are valuable tools to achieve that goal. While the topic addressed in this manuscript is of great scientific and public interest, I cannot recommend the publication of this study in its current form due to the reasons listed below. However, I believe that this paper has the potential to be a valuable addition to the scientific literature after thorough revisions.

1 Explanation of the physical mechanism of the emergent constraint

As already mentioned, I firmly believe that a solid physical mechanism behind every emergent constraint is vital to assess its plausibility. Therefore, I highly appreciate that the authors dedicated a large part of their paper on this physical mechanism in their section "Distinct synoptic cloud regimes associated with the extratropical low cloud variability in the high and low ECS models".

However, in its current state, the explanation of this physical mechanism is rather confusing. I had an especially hard time trying to understand the corresponding paragraph in the "Methods" section (1.412-434). Due to the missing references, I assume that this method is new, so it needs a much more detailed description. For example, the EOF analysis presented here deviates from the standard use in climate science where input data has dimensions time/latitude/longitude. Thus, please provide more details about your handling of the different dimensions (sub-region/pressure-level/time) here; if possible, with mathematical formulas. Is there some averaging involved or are some dimensions combined into one (like latitude/longitude in classical EOF analysis)? A more

detailed explanation why you think that EOF analysis is a suitable tool here and what you expect from it would also be most helpful.

Moreover, a little schematic that shows the different processes in the high- and low-ECS models (and observations) would be most helpful, in combination with a clear and concise description of these processes in the corresponding section. An example I have in mind is Fig. 4 of Watanabe et al. (2018). Right now, I have the impression that this information about the processes is scattered across different paragraphs within this section. In addition, given the length of this section, I am missing 1-2 sentences on this in the abstract. Given the importance of this physical mechanism of emergent constraints, I think it deserves more visibility in the abstract.

Finally, I think that the supporting figures used to explain this method are not really helpful in their current state. Within 6 lines (l.239-245) you mention Figs. S4, S5, and S6. All these figures are very busy and include lots of subplots. I tried to make sense of all of them for some time, but eventually gave up due to the massive amount of information they provide. If you really want show them, you need to add a detailed description for all of them. However, I strongly recommend to only focus on the important information by showing only a subset of these figures.

2 Statistical methods

I have large concerns regarding the description of the applied statistical methods here:

1. Whenever you show/mention errors (e.g., l.155, Fig, 2b and its caption) please always indicate what that error represents and how it was calculated, e.g., standard deviation, 5-95% range, etc. This is currently not mentioned anywhere in the manuscript.

2. Regarding the constrained ECS range you give in l.155: how did you calculate this posterior estimate? This needs to be specified in the manuscript since there are multiple different methods than can be used to compute this. See e.g., Brient (2020) or Schlund et al. (2020) for possible methods. However, given my comment in the subsequent section ("Suitability of observational data") it might be beneficial to simply stick to statements à la "high-ECS models are more consistent with the observations than low-ECS models" like you do in the abstract, and not give explicit ranges.

3. Whenever you talk about "significance" in a statistical sense (e.g., l.176,191), please indicate what you actually mean by that. Did you perform a statistical test for that? If yes, what's the null hypothesis? What's the p-value?

3 Suitability of observational data

Given the high sensitivity of your emergent constraint on the time period considered for the observations (l.357-375), I wonder if the observations you are using are really suited for that application and if your emergent constraint might be overconfident for that reason. The error bar you are showing seems to be fairly small given that high sensitivity; thus, it's absolutely crucial to indicate how that error bar was calculated (see comment in previous section). I find it also extremely worrisome that other observations do not show that strong seasonal variation in the extratropical LCF (l.282-289) and are thus more consistent with the low-ECS models, which is the exact opposite result of what you are showing in this study. If you believe that the observations you use are better and more trustworthy than others despite the deficiencies mentioned in l.357-375, you need to justify in detail why this is the case.

Moreover, I wonder if 6 years of data for the observations is enough to calculate a robust seasonal climatology. In contrast, you are using 35 years of data for the modes. I am also not convinced why observations from 2006-2011 can be used to constrain the model period 1980-2014 (which you are doing in Fig. 2b). It would be very helpful to add 1-2 sentences why you think this is the

case.

4 Technical problems with figures

1. Please show variable names and units next to the color bars of your figures, this makes it much easier to quickly see what's going on in a figure. Affected figures: Figs. 1, 2, 3, 6, S3, S4, S5, S6, S8, S9, S10.

2. Please add legends to all figures that need them. Affected figures: Figs. 4, 5, S5, S7, S8.

3. Please add missing axis labels (incl. variable name and units). Affected figures: Figs. 3, 6, S2, S3, S4, S5, S6, S7, S8, S9, S10.

4. Please use a sequential colormap for purely positive data (instead of a diverging colormap with blue and red colors, see Crameri et al. 2020 or <https://matplotlib.org/stable/tutorials/colors/colormaps.html>). Affected figures: Figs. 3, 6, S3, S4, S6, S8, S9, S10.

5. Please avoid using different Y-axis for the same quantity in Fig. 4. This is very misleading.

5 Additional aspects that should be considered

1. Given the high number of other studies that provide constraints on climate sensitivity based on cloud processes (e.g., Cesana et al. 2021, Myers et al. 2021, Tan et al. 2016, Zhai et al. 2015, emergent constraints discussed in Schlund et al. 2020), a discussion about the differences of these studies to your approach would be nice. Do you think that your emergent constraint is better? Why?

2. Given the recent debate about constraints that use the past warming trend like the one published by Tokarska et al. 2020 regarding the SST pattern effect (see e.g., Andrews et al. 2022) it could be helpful to mention that these constraints suffer from that problem. Do you think that your constraint is affected by that issue, too?

3. The ECS values you use from Schlund et al. (2020) are "effective climate sensitivity" values, which is only an approximation of the true equilibrium climate sensitivity and which might be biased (e.g., Rugenstein et al. 2020, Sanderson and Rugenstein 2022). It would be good to mention that limitation briefly in the manuscript.

Minor comments

1. Regarding the calculation of the shortwave cloud radiative effect (SWCRE) in this study: as far as I am aware, the CRE is usually calculated as the all-sky net (downwelling minus upwelling) top-of-atmosphere (TOA) radiation flux minus the clear-sky net TOA flux. Since the downwelling flux is identical for all-sky and clear-sky conditions, this results in $CRE = \text{clear-sky upwelling flux} - \text{all-sky upwelling flux}$ (see Stanfield et al. 2015, eq. (1) for this exact calculation, other relevant references are e.g., Grise and Medeiros 2016, Lauer et al. 2022, Fig. 7.7 of Boucher et al. 2013, Sec. 7.4.2.4.1 of Forster et al. 2021). This results in mostly negative values of SWCRE for the entire globe, corresponding to the cooling effect on the climate. In this study, you are using the opposite definition of SWCRE "defined by all-sky minus clear-sky TOA upwelling SW radiation" (1.562-563). While this certainly is not wrong and allows drawing identical conclusions (if used consistently), this unnecessarily complicates the comparison of results from this study with other literature. Thus, I strongly recommend to change the sign in the definition of SWCRE throughout the manuscript.

2. There are minor inaccuracies across the entire manuscript that can lead to misunderstandings. First, please always explicitly mention when you are considering only ocean grid cells in a specific region. For example, in l.142,675 (but also many other places), I am fairly sure you are talking about the ocean grid cells of the extratropics (given the text in the "Methods" section), not the entire extratropics. It would be very helpful to be more specific here. Second, whenever you refer to "surface temperature", please be more specific about that what you actually mean. For example, in l.151,401 (and other places) you are probably referring to "sea surface temperature", while in l.44,62 and SI l.40 you are talking about the "near-surface air temperature". Again, be more precise here.

Specific Comments

1. L.44,63: More specific: a doubling of the atmospheric CO₂ concentration.
2. L.67: This is incorrect. The inter-model range of CMIP5 is 2.1–4.7 K (see Fig. 1 and Tab. 1 of Meehl et al., 2020). The 1.5–4.5 K range is the "likely" range of ECS assessed by the IPCC AR5 using multiple lines of evidence (see Stocker et al., 2013).
3. L.72-74: It would be great if you could add a very brief explanation of the physical processes that lead to the connection "strong warming -> reduction of LCF" (similar to the explanation of the cloud phase feedback in l.82-84).
4. L.77-79: Please consider rephrasing this sentence. You are talking about a "shift in the reduction of TOA upwelling cloud SW radiation", but Fig. 1c shows the SWCRE, which is itself a difference of TOA upwelling radiations. My understanding from this sentence is that the maximum SWCRE difference is closer to the equator than the maximum LCF difference.
5. L.79-84: I am missing one or more references for these 2 sentences, e.g., McCoy et al. (2015).
6. L.93: Here you are talking about 18 model simulations, but the caption of Fig. 2 mentions 27 CMIP6 models. Which one is correct? Are you using a different number of models for the different panels of Fig. 2?
7. L.111-113: The definition of the "extratropics" and LCF should be mentioned earlier in the introduction (e.g., when it is first mentioned in l.74).
8. L.126-127: Please specify the "averaged over". From Fig. 2b I take that you are not averaging over the correlation coefficients of Fig. 2a but average over dLCF/dts first for each model and then correlate dLCF/dts vs. ECS across models, but this should be clearly stated.
9. L.132: You could calculate the correlation coefficient of these two correlation patterns to get a more quantitative statement than "greatly resembles".
10. L.134-137: You should mention here that you are referring to a correlation across CMIP models and the "range" is an inter-model range.
11. L.155-156: The sentence "[...] indicating that the high climate sensitivity with the ECS greater than 5K in several GCMs may be overestimated" seems to contradict your abstract which states that you find a "strong extratropical low cloud feedback that supports the high ECS models".
12. L.191-198: This entire paragraph about the short-term trend in the vertical velocity seems to be out of context as this is only fully discussed in detail in the subsequent section.
13. L. 267-270: I can only see the "significantly enhanced low clouds" in the CEMS2 and HadGEM3-LL models, but not in the CloudSat/CALIPSO observations.

14. L.302: "Fig. 5d" should probably be "Fig. 5e", "Fig. S7b4" should probably be "Fig. S7c4".
15. L. 306-307: Figs. 4a,c suggest this should probably be a "pronounced decrease with decreasing tropospheric ascending motion", not an "increase".
16. L.312-324: A relevant paper that should be mentioned in this paragraph is Grise and Medeiros (2016).
17. L.589-595: Please explain what the vertical line and shading represent in the caption of Fig. 2b.
18. L.404-407: It would be helpful to describe this calculation step by step or provide a formula. Currently the order in which the calculations described here (averaging over the corresponding grid cells, averaging over the periods 2061-2095/1980-2014, calculating the difference, calculation the fraction) are carried out is not 100% clear to me. Also, why don't you use the linear trend for the long-term change similarly to the seasonal and short-term time scale?
19. Fig. S5: Your contour lines show negative values for the cloud fraction. Can you explain why this is the case? Are you showing differences to here?

Technical Corrections

1. The usage of the symbol "dTS" in the denominator of dLCF/dts might be more appropriate than "dts".
2. The term "low cloud variability" might be a bit misleading. Instead of variability of low clouds, one could also think of low variability in clouds. Thus, I would recommend using a different expression for that.
3. L.112,394: I think a "cell" is missing after "grid".
4. L.589: LTS -> LCF.
5. Fig. 4: Elements in the legends are missing in Panels (a)-(e) (e.g., the colored lines are missing).

References

- Andrews, T., Bodas-Salcedo, A., Gregory, J. M., Dong, Y., Armour, K. C., Paynter, D., et al. (2022). On the effect of historical SST patterns on radiative feedback. *Journal of Geophysical Research: Atmospheres*, 127, e2022JD036675. <https://doi.org/10.1029/2022JD036675>.
- Boucher, O., D. Randall, P. Artaxo, C. Bretherton, G. Feingold, P. Forster, V.-M. Kerminen, Y. Kondo, H. Liao, U. Lohmann, P. Rasch, S.K. Satheesh, S. Sherwood, B. Stevens and X.Y. Zhang, 2013: Clouds and Aerosols. In: *Climate Change 2013: The Physical Science Basis. Contribution of Working Group I to the Fifth Assessment Report of the Intergovernmental Panel on Climate Change* [Stocker, T.F., D. Qin, G.-K. Plattner, M. Tignor, S.K. Allen, J. Boschung, A. Nauels, Y. Xia, V. Bex and P.M. Midgley (eds.)]. Cambridge University Press, Cambridge, United Kingdom and New York, NY, USA.
- Brient, F. Reducing Uncertainties in Climate Projections with Emergent Constraints: Concepts, Examples and Prospects. *Adv. Atmos. Sci.* 37, 1–15 (2020). <https://doi.org/10.1007/s00376-019-9140-8>.
- Cesana, G. V. & Del Genio, A. D. Observational constraint on cloud feedbacks suggests moderate

climate sensitivity. *Nature Climate Change* 11, 213-218, <https://doi.org/10.1038/s41558-020-00970-y> (2021).

Cramer, F., Shephard, G.E. & Heron, P.J. The misuse of colour in science communication. *Nat Commun* 11, 5444 (2020). <https://doi.org/10.1038/s41467-020-19160-7>.

Forster, P., Storelvmo, K., Armour, W., Collins, J.-L., Dufresne, D., Frame, D.J., Lunt, T., Mauritsen, M.D., Palmer, M., Watanabe, M., Wild, and H. Zhang, 2021: The Earth's Energy Budget, Climate Feedbacks, and Climate Sensitivity. In *Climate Change 2021: The Physical Science Basis. Contribution of Working Group I to the Sixth Assessment Report of the Intergovernmental Panel on Climate Change* [Masson-Delmotte, V., P. Zhai, A. Pirani, S.L. Connors, C. Péan, S. Berger, N. Caud, Y. Chen, L. Goldfarb, M.I. Gomis, M. Huang, K. Leitzell, E. Lonnoy, J.B.R. Matthews, T.K. Maycock, T. Waterfield, O. Yelekçi, R. Yu, and B. Zhou (eds.)]. Cambridge University Press, Cambridge, United Kingdom and New York, NY, USA, pp. 923–1054, <https://doi.org/10.1017/9781009157896.009>.

Grise, Kevin M., and Brian Medeiros (2016). "Understanding the varied influence of midlatitude jet position on clouds and cloud radiative effects in observations and global climate models." *Journal of Climate* 29.24: 9005-9025, <https://doi.org/10.1175/JCLI-D-16-0295.1>.

Lauer, A., Bock, L., Hassler, B., Schröder, M., & Stengel, M. (2023). Cloud Climatologies from Global Climate Models—A Comparison of CMIP5 and CMIP6 Models with Satellite Data, *Journal of Climate*, 36(2), 281-311, <https://doi.org/10.1175/JCLI-D-22-0181.1>.

McCoy, D. T., Hartmann, D. L., Zelinka, M. D., Ceppi, P., & Grosvenor, D. P. (2015). Mixed-phase cloud physics and Southern Ocean cloud feedback in climate models. *Journal of Geophysical Research: Atmospheres*, 120(18), 9539–9554. <https://doi.org/10.1002/2015jd023603>.

Meehl, G. A., Senior, C. A., Eyring, V., Flato, G., Lamarque, J.-F., Stouffer, R. J., Taylor, K. E., & Schlund, M. (2020). Context for interpreting equilibrium climate sensitivity and transient climate response from the CMIP6 Earth system models. *Science Advances*, 6(26), eaba1981. <https://doi.org/10.1126/sciadv.aba1981>.

Myers, T. A. et al. Observational constraints on low cloud feedback reduce uncertainty of climate sensitivity. *Nature Climate Change* 11, 501-507, <https://doi.org/10.1038/s41558-021-01039-0> (2021).

Rugenstein, M., Bloch-Johnson, J., Gregory, J., Andrews, T., Mauritsen, T., Li, C., et al. (2020). Equilibrium climate sensitivity estimated by equilibrating climate models. *Geophysical Research Letters*, 47, e2019GL083898. <https://doi.org/10.1029/2019GL083898>.

Sanderson, B. M. and Rugenstein, M.: Potential for bias in effective climate sensitivity from state-dependent energetic imbalance, *Earth Syst. Dynam.*, 13, 1715–1736, <https://doi.org/10.5194/esd-13-1715-2022>, 2022.

Schlund, M., Lauer, A., Gentine, P., Sherwood, S. C., and Eyring, V.: Emergent constraints on equilibrium climate sensitivity in CMIP5: do they hold for CMIP6?, *Earth Syst. Dynam.*, 11, 1233–1258, <https://doi.org/10.5194/esd-11-1233-2020>, 2020.

Stanfield, R. E., Dong, X., Xi, B., Del Genio, A. D., Minnis, P., Doelling, D., & Loeb, N. (2015). Assessment of NASA GISS CMIP5 and Post-CMIP5 Simulated Clouds and TOA Radiation Budgets Using Satellite Observations. Part II: TOA Radiation Budget and CREs, *Journal of Climate*, 28(5), 1842-1864, <https://doi.org/10.1175/JCLI-D-14-00249.1>.

Stocker, T. F., Qin, D., Plattner, G.-K., Alexander, L. V., Allen, S. K., Bindoff, N. L., Bréon, F.-M., Church, J. A., Cubasch, U., Emori, S., Forster, P., Friedlingstein, P., Gillett, N., Gregory, J. M., Hartmann, D. L., Jansen, E., Kirtman, B., Knutti, R., Kumar, K. K.,... Xie, S.-P. (2013). *Technical Summary*. Cambridge University Press.

Tan, I., Storelvmo, T., & Zelinka, M. D. (2016). Observational constraints on mixed-phase clouds imply higher climate sensitivity. *Science*, 352(6282), 224-227, <https://doi.org/10.1126/science.aad5300>.

Tokarska, K. B. et al. Past warming trend constrains future warming in CMIP6 models. *Science Advances* 6, eaaz9549, <https://doi.org/10.1126/sciadv.aaz9549> (2020).

Watanabe, M., Kamae, Y., Shiogama, H. et al. Low clouds link equilibrium climate sensitivity to hydrological sensitivity. *Nature Clim Change* 8, 901–906 (2018). <https://doi.org/10.1038/s41558-018-0272-0>.

Zhai, C., Jiang, J. H. & Su, H. Long-term cloud change imprinted in seasonal cloud variation: More evidence of high climate sensitivity. *Geophys. Res. Lett.* 42, 8729-8737, <https://doi.org/10.1002/2015GL065911> (2015).

Reviewer #2 (Remarks to the Author):

Jiang et al. present an argument for a new emergent constraint that relates the spread in climate-model projections of equilibrium climate sensitivity (ECS) to the spread in climate-model estimates of the seasonal cycle of extratropical low-level clouds. The research topic is relevant to a broad scientific audience, and the writing and figures are clear. However, I believe that the paper lacks a convincing physical explanation for the emergent constraint, and it does not sufficiently discuss how the results relate to the existing literature. The method of uncertainty quantification and choice of observational data for the emergent constraint are also not sufficiently explained and justified. If these issues are addressed, then I believe that the paper may be suitable for publication. I recommend major revision.

General Comments

- One of the most important shortcomings of the paper is the lack of a convincing physical explanation for the emergent constraint. The authors state that the seasonal cycle of extratropical storm-track clouds is linked to ECS, but they offer no specific physical argument for why this should be the case. Why does the seasonal variation of storm-track clouds provide predictive skill for how these clouds will change in a warming world? I would expect CO₂-driven warming and seasonal changes in insolation to cause different equator-to-pole temperature gradients and static stability changes, and therefore different variations in baroclinicity and storm-track activity. I think the authors need to explain this better. Furthermore, the authors base their arguments on relationships between clouds and local surface temperature, but the storm track is more sensitive to changes in equator-to-pole temperature gradients than local surface temperature. Thus, there seems to be an inconsistency between the interpretation that the authors provide and the relationships they examine. Please explain this apparent inconsistency.
- One key lesson we learned from CMIP6 is that many of the emergent constraints developed from CMIP5 models do not hold up to scrutiny when applied to the CMIP6 ensemble. Now that we know this, I think it is important that any new emergent constraints are tested across both CMIP5 and CMIP6 models. I recommend that the authors examine if their emergent constraint holds true in the CMIP5 ensemble. If so, this would make the argument more convincing.
- The paper would be stronger if the authors discuss how their results compare with the existing literature in more detail. For example, Grise and Polvani (2016) found weak or insignificant relationships between ECS and poleward shifts of the extratropical jet streams in a warming world across GCMs. Ceppi and Hartmann (2015) found a weak relationship between natural variations in latitude of the Southern Hemisphere extratropical jet and SW cloud radiative effects in observations. These findings seem to contradict the proposed emergent constraint. Please explain these apparent discrepancies.
- The uncertainty quantification needs to be described. In particular, how is spatial autocorrelation between the grid boxes accounted for? If you assume that every grid box is independent, then you will underestimate uncertainty. I recommend accounting for spatial autocorrelation following a method similar to that of Myers et al. (2021). Also, the authors show that their observational quantity that determines the emergent constraint (dLCF/dts) depends on the choice of observational dataset used. However, the authors only show one of the observational datasets in

the vertical line in Fig. 2b. Why not show all of the results? Without showing all results or explaining why you picked just one, you risk sounding like you are selectively choosing evidence that produces the highest ECS, which would be problematic. Please justify your choices more clearly in the text.

Specific Comments

- Line 76: "generate" -> "generates"
- Line 77: "to amplify" -> "that amplifies"
- Line 155: please state the confidence level for the ECS uncertainty range (one sigma, 90% CI, 95% CI, etc.)
- Line 189: the first CERES measurements are from 1997 from the TRMM satellite. The model trends are computed starting in 1980, which is much earlier. It would be good to revise this sentence so it doesn't sound like you are overstating the degree of agreement between the CERES trends and the model trends.
- Fig 4 (a-d): This figure would be clearer if you plot all of the values in each panel on the same y-axis. (i.e. all of the LCF data in Fig. 4a are plotted with the same y axis, all of the TS data in Fig. 4b are plotted with the same y axis, etc.)
- Line 196: The same criticisms you make about biases in the vertical velocity data from reanalysis apply to the vertical velocity data from climate models. I recommend rephrasing this sentence so that it doesn't imply that biases in reanalysis vertical velocity are worse than biases in global climate models, which is not true.
- Line 218: It would be clearer if you could specifically state the large-scale cloud-controlling factors that are similar between the high-ECS and low-ECS models so that the reader doesn't have to check the supporting information.
- Fig. 5: It would be clearer to plot (a), (c), and (e) on the same y-axis scale, or least use the same range of values in the y axis of each panel (e.g. y axis in (a) ranging from 45 to 70, y axis in (c) ranging from 25 to 50, etc.). Also, please explain why you chose to plot EOF mode 2 in the top row, EOF mode 2 and 3 in the middle row, and EOF mode 3 in the bottom row. This seems unnecessarily complicated and makes it difficult to compare across the rows.

References

Ceppi, P., Hartmann, D.L. Connections Between Clouds, Radiation, and Midlatitude Dynamics: a Review. *Curr Clim Change Rep* 1, 94–102 (2015)

Grise, K. M., and Polvani, L. M. (2016), Is climate sensitivity related to dynamical sensitivity?, *J. Geophys. Res. Atmos.*, 121, 5159– 5176, doi:10.1002/2015JD024687.

Myers, T.A., Scott, R.C., Zelinka, M.D. et al. Observational constraints on low cloud feedback reduce uncertainty of climate sensitivity. *Nat. Clim. Chang.* 11, 501–507 (2021).
<https://doi.org/10.1038/s41558-021-01039-0>

We thank the two reviewers for insightful comments and suggestions on an earlier version of this manuscript. The following are our point-by-point responses. The review comments are shown in black with our responses in blue.

Responses to Reviewer #1:

1 Explanation of the physical mechanism of the emergent constraint

As already mentioned, I firmly believe that a solid physical mechanism behind every emergent constraint is vital to assess its plausibility. Therefore, I highly appreciate that the authors dedicated a large part of their paper on this physical mechanism in their section "Distinct synoptic cloud regimes associated with the extratropical low cloud variability in the high and low ECS models".

However, in its current state, the explanation of this physical mechanism is rather confusing. I had an especially hard time trying to understand the corresponding paragraph in the "Methods" section (l.412-434). Due to the missing references, I assume that this method is new, so it needs a much more detailed description. For example, the EOF analysis presented here deviates from the standard use in climate science where input data has dimensions time/latitude/longitude. Thus, please provide more details about your handling of the different dimensions (sub-region/pressure-level/time) here; if possible, with mathematical formulas. Is there some averaging involved or are some dimensions combined into one (like latitude/longitude in classical EOF analysis)? A more detailed explanation why you think that EOF analysis is a suitable tool here and what you expect from it would also be most helpful.

We thank this reviewer for the very detailed comments and constructive suggestions. The EOF approach to identify the leading cloud regimes over the extratropics developed in this study is indeed different from previous studies. Following this reviewer's suggestion, we have made significant improvement in the "Methods" part by providing as much details as we can, particularly on the approach to construct the covariance matrix for the EOF analysis, including its spatial and temporal points. The reason that we use EOF to identify the leading cloud regimes is that it is simple while objective to identify the leading vertical cloud modes. Since different models share very similar leading cloud regimes, and the simulated seasonal and interannual variability of vertical clouds in the models can be well reproduced just using the three leading modes, it makes this approach very useful to identify distinct leading cloud regimes controlling the extratropical LCF variability in the high and low ECS models.

Please see the modified "Cloud regime analysis" section under the "Methods" (Lines 438-462).

Moreover, a little schematic that shows the different processes in the high- and low-ECS models (and observations) would be most helpful, in combination with a clear and concise description of these processes in the corresponding section. An example I have in mind is Fig. 4 of Watanabe et al. (2018). Right now, I have the impression that this information about the processes is scattered across different paragraphs within this section. In addition, given the length of this section, I am missing 1-2 sentences on this in the abstract. Given the importance of this physical mechanism of

emergent constraints, I think it deserves more visibility in the abstract.

We thank this reviewer for this nice suggestion. After careful considerations, we felt it a bit challenging to make a nice schematic to summarize the distinct cloud feedback processes between the high and low ECS models. Instead, we have made significant efforts to improve the description of the physical processes underlying the distinct extratropical low-cloud feedback in the high and low ECS models. For example, a summary of these processes is provided in the “Summary and discussions” part (Lines 323-335).

(Lines 323-335) “In the high ECS models, the LCF variability tends to be dominated by a mid-top or a combined mid- and low-top cloud regime that is strongly coupled to the lower-to-mid-tropospheric vertical velocity. During the seasonal migration from winter to summer, tropospheric ascending motion over the extra-tropics weakens due to reduced baroclinicity and storm-track variability, and leads to a significant reduction of extratropical LCF in these models. Under a warming climate, weakening of the ascending motion over the extra-tropics, possibly associated with the expansion of the Hadley Cell^{47,48,49,50} and/or the poleward shift of mid-latitude jet stream^{45,51}, can also lead to reduced extratropical LCF in the high ECS models through modulation of the vertically-extended clouds, therefore, a positive low-cloud feedback that leads to rapid warming over the extratropics in these models. In contrast, in the low ECS models as indicated by MPI-ESM2-LR, the variability of extratropical low clouds is mainly dominated by a low-top cloud regime, which exhibits a seasonal cycle that largely follows the lower-tropospheric stability, i.e., with a maximum in summer and a minimum in winter, and lacks a significant trend under a warming climate.”

As suggested by this reviewer, the main difference in the cloud regimes between the high and low ECS models is also very briefly discussed in the abstract in the revised version due to the word limit in abstract (Lines 51-53).

(Lines 51-53) “The distinct seasonality in extratropical LCF in climate models is ascribed to their different prevailing cloud regimes governing the extratropical LCF variability.”

Finally, I think that the supporting figures used to explain this method are not really helpful in their current state. Within 6 lines (l.239-245) you mention Figs. S4, S5, and S6. All these figures are very busy and include lots of subplots. I tried to make sense of all of them for some time, but eventually gave up due to the massive amount of information they provide. If you really want show them, you need to add a detailed description for all of them. However, I strongly recommend to only focus on the important information by showing only a subset of these figures.

Following this reviewer’s suggestion, in the revised manuscript we have significantly modified the supporting figures, including:

- 1) We removed the old Fig. S8 on EOF analyses of daily vertical cloud profiles based on CloudSat/CALIPSO observations considering significant sampling issues involved with the daily CloudSat/CALIPSO data.

- 2) We removed the old Figs. S9-S10 on comparison between the CloudSat/CALIPSO and CALIPSO-only cloud profiles since the advantage of the combined CloudSat/CALIPSO dataset for studies on extratropical clouds has been well known (Haynes et al. 2011·Govekar et al. 2011·Naud et al. 2019·Li et al. 2014). We feel that these kind of details are not necessary for the general readers who are interested in the model simulated climate sensitivity.
- 3) We also removed the old Fig. S7 on reconstruction of the interannual LCF variability during the period of 1980-2014 in three model simulations considering that was a very busy plot as this reviewer mentioned and the Fig. 5 (panels b,d,f) well serves the purpose of the old Fig. S7;
- 4) While we still keep the old Fig. S6 (now the Supplementary Fig. 5) to illustrate how well the three leading EOF modes can depict the seasonal cycle of vertical cloud variability in the three GCMs, we significantly reduced figure panels by only showing the vertical-temporal profiles of total cloud fraction (column 1), total cloud fraction explained by the long-term climatology plus variability associated with the three leading modes (column 2), and variability of cloud fraction anomalies associated with the three leading modes (columns 3-5). We believe these changes will make this figure much more readable while still provide detailed information on how the reconstruction using the three leading EOF modes works, and the different dominant cloud regimes in these models.

2 Statistical methods

I have large concerns regarding the description of the applied statistical methods here:

1. Whenever you show/mention errors (e.g., 1.155, Fig. 2b and its caption) please always indicate what that error represents and how it was calculated, e.g., standard deviation, 5-95% range, etc. This is currently not mentioned anywhere in the manuscript.

Thanks for these nice suggestions. Following this reviewer's suggestion below, in the revised manuscript we decided to avoid providing a range of the estimated ECS using the constraint from the observed seasonal dLCF/dTS. Indeed, rather than predicting a reasonable range of the ECS, the key point of this study is to point out the significant model deficiencies in the low ECS in representing the seasonal cycle of extratropical low clouds, which can be closely associated with model biases in depicting cloud feedback processes over the extratropics. Meanwhile, uncertainties associated with the observed dLCF/dTS and linear regression in Fig. 2b have been added in the figure caption, which were overlooked in the original manuscript.

(Lines 672-677) "The light grey shaded areas around the regression line in b) represent the standard prediction errors by the linear fit following Schlund et al. (2020)²⁵. The vertical red line in b) corresponds to the seasonal dLCF/dTS derived from the climatological seasonal cycle of LCF and TS based on the satellite observations for the period of 2006-2011, with its uncertainty (dark grey shading) estimated by the mean and one standard deviation of the observed seasonal dLCF/dTS in each year of 2006-2011. See Methods for details in deriving the dLCF/dTS on various time scales based on both models and observations. "

In the same time, more detailed discussions on the uncertainties associated with the dLCF/dTS have also been added in the “Observational datasets” part under “Methods”.

(Lines 421-426) “While a seasonal dLCF/dTS of $-1.71\%/K$ over the extratropics from the CloudSat/CALIPSO observations is derived based on climatological seasonal cycle of vertical clouds (Fig. 2b, vertical red line), its year-to-year variability is relatively weak with a mean value of $-1.65\%/K$ and a standard deviation of $0.27\%/K$ during the six-year period. This mean value of the seasonal dLCF/dTS along with the one standard deviation value is used to estimate uncertainties involved with the observed seasonal dLCF/dTS (see the vertical grey bar in Fig. 2b).”

2. Regarding the constrained ECS range you give in l.155: how did you calculate this posterior estimate? This needs to be specified in the manuscript since there are multiple different methods than can be used to compute this. See e.g., Brient (2020) or Schlund et al. (2020) for possible methods. However, given my comment in the subsequent section (“Suitability of observational data”) it might be beneficial to simply stick to statements à la “high-ECS models are more consistent with the observations than low-ECS models” like you do in the abstract, and not give explicit ranges.

As mentioned above, follow this reviewer’s suggestion, in the revised version we decided to not provide an explicit ECS range based on the emergent constraint. This sentence has been removed in the revised manuscript.

3. Whenever you talk about “significance” in a statistical sense (e.g., l.176,191), please indicate what you actually mean by that. Did you perform a statistical test for that? If yes, what's the null hypothesis? What's the p-value?

In the revised manuscript, we added p-values along with the correlation coefficients in most places particularly with moderate correlations. With a null hypothesis that there is no correlation between two variables, it will be considered a significant correlation when a p-value is smaller than 0.05.

3 Suitability of observational data

Given the high sensitivity of your emergent constraint on the time period considered for the observations (l.357-375), I wonder if the observations you are using are really suited for that application and if your emergent constraint might be overconfident for that reason. The error bar you are showing seems to be fairly small given that high sensitivity; thus, it's absolutely crucial to indicate how that error bar was calculated (see comment in previous section). I find it also extremely worrisome that other observations do not show that strong seasonal variation in the extratropical LCF (l.282-289) and are thus more consistent with the low-ECS models, which is the exact opposite result of what you are showing in this study. If you believe that the observations you use are better and more trustworthy than others despite the deficiencies mentioned in l.357-375, you need to justify in detail why this is the case.

Regarding the selection of the observational dataset to derive the seasonal dLCF/dTS, it has been widely reported and well-known that the CloudSat/CALIPSO combined cloud product (2B-GEOPROFLIDAR) is the best dataset in depicting the clouds associated with the extratropical cyclones (Naud et al. 2010; Govekar et al. 2011; Li et al. 2014; Mace and Zhang 2014). In the original manuscript, the reason that we mentioned the different values of the seasonal dLCF/dTS derived based on the CloudSat/CALIPSO data between 2006-2011 when it was fully functioning and 2012-2018 when it was in a degraded mode after a serious battery anomaly in 2011, and between CloudSat/CALIPSO and CALIPSO-only was to justify the employment of the CloudSat/CALIPSO data during the 2006-2011 for this study. As shown in our original manuscript, even the active LIDAR observations of vertical clouds by CALIPSO failed in detecting the vertically-extended extratropical clouds (see Fig. A1 below), other passive sensors such as ISCCP will have more serious deficiencies in depicting the low clouds associated with the vertically-extended storm-track clouds due to their limitations in detecting low clouds collocated with deep or multi-layered clouds, as illustrated in (Bodas-Salcedo et al. 2014). The passive sensing also has difficulties in accurately detecting the cloud-top height (Bodas-Salcedo et al. 2014). Therefore, there is no doubt the combined CloudSat/CALIPSO dataset provides the best observations to derive the dLCF/dTS in this study.

Fig. A1 Panel (a) shows time evolution of vertical cloud fractions over the region of 90-120°E; 55-45°S during a one-year period from the Combined CloudSat/CALIPSO observations. Panel (b) is the same as (a) but shows vertical cloud evolution during the same period only detected by CALIPSO. White strips denote missing data in observations.

Considering that a paper to be published at Nature Communications will be mainly targeted at general readers who not necessarily need to know such detailed information regarding the uncertainties involved with different satellite sensors, and to concentrate this paper mainly on scientific discussions, in this revised manuscript, we decided to remove very detailed discussions on observational uncertainties in the main text. Instead, we provide a brief introduction of the combined CloudSat/CALIPSO cloud observations in “Methods”, why it is the best dataset available for vertical cloud observations over other datasets, particularly for studies on clouds associated with extratropical cyclones.

Corresponding modifications on this part include:

(Lines 413-418) “The observational dataset for vertical cloud profiles is based on a combined product from the CloudSat and Cloud-Aerosol Lidar and Infrared Pathfinder Satellite Observation (CALIPSO) satellites (2B-GEOPROFLIDAR; version P2R05)²⁸. The combined CloudSat/CALIPSO dataset has been considered the best satellite observations for vertical cloud structures associated with extratropical cyclones^{55,36,56,43}, mainly due to its advantage of the Cloud Profiling Radar (CPR) aboard CloudSat in detecting optically thick hydrometeor layers, and the CALIPSO lidar in detecting optically thin cloud layers that could be missed by the CPR^{57,58,28}. ”

(Lines 430-436) “It is also noteworthy that global low-cloud observations are also provided by several passive-sensing satellites, such as the International Satellite Cloud Climatology Project (Rossow et al. 2022). While these passive instruments provide useful information of horizontal distribution of low-top clouds, they have limitations in detecting low clouds collocated with deep or multi-layered clouds, such as those associated with extratropical cyclones along the mid-latitude storm tracks. The passive sensing also has difficulties in accurately detecting the cloud-top height (Bodas-Salcedo et al. 2014). Therefore, the low-cloud fractions based on these passive-sensing satellite observations are not supposedly to be directly compared to those derived from the CloudSat/CALIPSO observations and model simulations in this study.”

Also, please refer to our previous responses on estimates of observational uncertainties.

Hope it reads better in the revised version.

Moreover, I wonder if 6 years of data for the observations is enough to calculate a robust seasonal climatology. In contrast, you are using 35 years of data for the modes. I am also not convinced why observations from 2006-2011 can be used to constrain the model period 1980-2014 (which you are doing in Fig. 2b). It would be very helpful to add 1-2 sentences why you think this is the case.

This is a very good point. To make sure a 6-year period is enough to get a stable climatological seasonal cycle, we also calculated seasonal evolution of clouds over the extratropics in model simulations but only based on 2006-2011. As shown below (Fig. A2), the results are largely identical to those based on the entire 35-year period of model data in Fig. 3.

Also, as discussed previously and mentioned in details in “Methods”, during the 6-year period of 2006-2011, the year-to-year variability in the seasonal dLCF/dTS based on CloudSat/CALIPSO is also relatively small, with a mean value of -1.65%/K and a standard deviation of 0.27%/K. This suggests the seasonal dLCF/dTS over the extratropics as derived in this study is very stable.

Based on these, we feel confident to use the dLCF/dTS derived from the 6-year CloudSat/CALIPSO observations to validate model simulations.

Fig. A2 Same as in Fig. 3, but based on model output for the period of 2006-2011.

4 Technical problems with figures

1. Please show variable names and units next to the color bars of your figures, this makes it much easier to quickly see what's going on in a figure. Affected figures: Figs. 1, 2, 3, 6, S3, S4, S5, S6, S8, S9, S10.

Following this reviewer’s suggestion, variable names along with units have been labeled next to the color bars in these figures. Note that several plots (old Figs. S7-S10) have been removed in this revised version.

2. Please add legends to all figures that need them. Affected figures: Figs. 4, 5, S5, S7, S8.

Legends have been added in these figures.

3. Please add missing axis labels (incl. variable name and units). Affected figures: Figs. 3, 6, S2, S3, S4, S5, S6, S7, S8, S9, S10.

Axis labels have been added for these figures.

4. Please use a sequential colormap for purely positive data (instead of a diverging colormap with blue and red colors, see Crameri et al. 2020 or <https://matplotlib.org/stable/tutorials/colors/colormaps.html>). Affected figures: Figs. 3, 6, S3, S4, S6, S8, S9, S10.

Thanks for this suggestion. A sequential color-map has been applied for these plots except Fig. S4, which is for anomalous vertical velocity with both negative and positive values.

5. Please avoid using different Y-axis for the same quantity in Fig. 4. This is very misleading.

This has been fixed by only showing anomalies of each variable with its long-term climatology removed.

5 Additional aspects that should be considered

1. Given the high number of other studies that provide constraints on climate sensitivity based on cloud processes (e.g., Cesana et al. 2021, Myers et al. 2021, Tan et al. 2016, Zhai et al. 2015, emergent constraints discussed in Schlund et al. 2020), a discussion about the differences of these studies to your approach would be nice. Do you think that your emergent constraint is better? Why?

Thanks for these nice suggestions. The emergent constraint proposed in this study and those in previous studies mentioned above focused on feedback processes over different regions. While our constraint provides a unique metric for extratropical low-cloud feedback, it is not necessarily better than others.

We have added some brief discussions based on these comments (Lines 376-383):

(Lines 376-383) “We also note that a model’s overall climate sensitivity can be contributed by multiple feedback processes over the globe in addition to the extratropical low-cloud feedback examined in this study (Schlund et al. 2020, Sherwood et al. 2020, Zelinka et al. 2022). For example, several other cloud-based emergent metrics have been recently proposed to constrain model climate sensitivity, including tropical shallow cumulus and stratocumulus clouds (Cesana and Del Genio 2021), seasonal cycle of subtropical marine LCF (Zhai et al. 2015), or the regime-averaged marine low-cloud feedbacks between 60°S-60°N (Myers et al. 2021). The emergent relationship between model ECS and the seasonal variability of extratropical LCF proposed in this study provides a unique metric to quantify model representation of low-cloud feedbacks over the extratropics.”

2. Given the recent debate about constraints that use the past warming trend like the one published by Tokarska et al. 2020 regarding the SST pattern effect (see e.g., Andrews et al. 2022) it could be helpful to mention that these constraints suffer from that problem. Do you think

that your constraint is affected by that issue, too?

This is another very good point. As previously discussed, the year-to-year variability of the seasonal dLCF/dTS based on CloudSat/CALIPSO during the 2006-2011 is relatively weak with a mean of -1.65 %/K and a standard deviations of 0.27%/K (vertical shading in Fig. 2). Also the climatological seasonal cycle of extratropical clouds based on 6-year model data is largely identical to that based on the 35-year period (see Fig. A2 and compared to Fig. 3). Based on these results, we consider that our constraint based on the extratropical seasonal dLCF/dTS is not subject to the SST pattern effect. Since the SST pattern effect is mainly involved with tropical/subtropical region, we decided not to include these discussions in the revised manuscript.

3. The ECS values you use from Schlund et al. (2020) are "effective climate sensitivity" values, which is only an approximation of the true equilibrium climate sensitivity and which might be biased (e.g., Rugenstein et al. 2020, Sanderson and Rugenstein 2022). It would be good to mention that limitation briefly in the manuscript.

Thanks for pointing out this. Considering that this "effective climate sensitivity" approach to estimate ECS in model simulations has been widely used in the community, we decided not to particularly mention this in the main text to make the paper read more smoothly. Instead, these has been mentioned in Fig. 2 captions, also supporting Table 1 when introducing the ECS values from the CMIP6 models.

For example, Lines (678-682): "Note that the ECS value predicted by each model used in this study is the "effective climate sensitivity" from Schlund et al. (2020)²⁵, an approximation of the equilibrium climate sensitivity that might be biased^{60,61}."

Minor comments

1. Regarding the calculation of the shortwave cloud radiative effect (SWCRE) in this study: as far as I am aware, the CRE is usually calculated as the all-sky net (downwelling minus upwelling) top-of-atmosphere (TOA) radiation flux minus the clear-sky net TOA flux. Since the downwelling flux is identical for all-sky and clear-sky conditions, this results in CRE = clear-sky upwelling flux minus all-sky upwelling flux (see Stanfield et al. 2015, eq. (1) for this exact calculation, other relevant references are e.g., Grise and Medeiros 2016, Lauer et al. 2022, Fig. 7.7 of Boucher et al. 2013, Sec. 7.4.2.4.1 of Forster et al. 2021). This results in mostly negative values of SWCRE for the entire globe, corresponding to the cooling effect on the climate. In this study, you are using the opposite definition of SWCRE "defined by all-sky minus clear-sky TOA upwelling SW radiation" (1.562-563). While this certainly is not wrong and allows drawing identical conclusions (if used consistently), this unnecessarily complicates the comparison of results from this study with other literature. Thus, I strongly recommend to change the sign in the definition of SWCRE throughout the manuscript.

We appreciate this reviewer's detailed explanation of SWCRE. This has been fixed in the revised manuscript which involves with Figs. 1 and 4.

2. There are minor inaccuracies across the entire manuscript that can lead to misunderstandings. First, please always explicitly mention when you are considering only ocean grid cells in a specific region. For example, in l.142,675 (but also many other places), I am fairly sure you are talking about the ocean grid cells of the extratropics (given the text in the "Methods" section), not the entire extratropics. It would be very helpful to be more specific here. Second, whenever you refer to "surface temperature", please be more specific about that what you actually mean. For example, in l.151,401 (and other places) you are probably referring to "sea surface temperature", while in l.44,62 and SI l.40 you are talking about the "near-surface air temperature". Again, be more precise here.

In the revised version, we explicitly mentioned that the analyses were applied to ocean grid points only:

(Lines 70-71) "all analyses in this study are conducted over ocean grid points to avoid even more complicated feedback processes over the land.."

We also explicitly defined surface temperature in the revised paper:

(Lines 71-73) "...and surface temperature is referred to skin temperature if not specifically defined, i.e., sea surface temperature over ocean grid cells."

"near-surface air temperature" are specifically defined in Lines 44, 60, and captions for Table S1.

Specific Comments

1. L.44,63: More specific: a doubling of the atmospheric CO₂ concentration.

Modified as suggested.

2. L.67: This is incorrect. The inter-model range of CMIP5 is 2.1–4.7 K (see Fig. 1 and Tab. 1 of Meehl et al., 2020). The 1.5–4.5 K range is the "likely" range of ECS assessed by the IPCC AR5 using multiple lines of evidence (see Stocker et al., 2013).

Thanks for the correction. This has been modified.

3. L.72-74: It would be great if you could add a very brief explanation of the physical processes that lead to the connection "strong warming -> reduction of LCF" (similar to the explanation of the cloud phase feedback in l.82-84).

A brief discussion on this has been added:

(Lines 80-83) "While reduced low clouds with surface warming over tropics or subtropics has been ascribed to a drying effect in the boundary layer due to enhanced vertical moisture gradient and thus mixing^{12,13,14}, processes underlying reduced extratropical low clouds under a

warming climate remain poorly understood, and are highly relevant to the findings in this study.”

4. L.77-79: Please consider rephrasing this sentence. You are talking about a "shift in the reduction of TOA upwelling cloud SW radiation", but Fig. 1c shows the SWCRE, which is itself a difference of TOA upwelling radiations. My understanding from this sentence is that the maximum SWCRE difference is closer to the equator than the maximum LCF difference.

This has been modified as follows:

(Lines 84-85) “A slightly equatorward shift in the maximum TOA SWCRE changes relative to the maximum reduction of LCF is noted over 30-60°N/S (c.f. Fig. 1c, 1e).”

5. L.79-84: I am missing one or more references for these 2 sentences, e.g., McCoy et al. (2015).

This reference has been included in the revised paper:

(Lines 86-87) “...these equatorward shifts in SWCRE can also be due to a negative feedback involved with mixed-phase clouds over the higher latitudes of the extra-tropics (Zelinka et al. 2012; McCoy et al. 2015) ”

6. L.93: Here you are talking about 18 model simulations, but the caption of Fig. 2 mentions 27 CMIP6 models. Which one is correct? Are you using a different number of models for the different panels of Fig. 2?

There are only 18 GCMs available for the dLCF/dTS associated with the long-term trend. This has been explicitly mentioned in the text and caption of Fig. 2 (Lines 99, 671-672).

(Line 99) “...exhibits a high negative correlation (~ -0.81) with the model ECS across 18 available GCM simulations”

(Lines 671-672) “Note that only 18 GCMs are available for the dLCF/dTS associated with the long-term trend in c) and d).”

7. L.111-113: The definition of the "extratropics" and LCF should be mentioned earlier in the introduction (e.g., when it is first mentioned in l.74).

Thanks. These have been moved to the “introduction” part in the revised paper.

(Lines 69-70) “Hereafter, if not stated otherwise, the extra-tropics is referred to the latitude belts of 60°S-30°S and 30°N-60°N.”

(Lines 73-74) “Low cloud fraction (LCF) in both models and observations is derived by vertical cloud fractions below 700hPa using a maximum overlapping assumption.”

8. L.126-127: Please specify the "averaged over". From Fig. 2b I take that you are not averaging

over the correlation coefficients of Fig. 2a but average over dLCF/dts first for each model and then correlate dLCF/dts vs. ECS across models, but this should be clearly stated.

This reviewer is correct. We averaged dLCF/dTS over the extratropics for each model first and then calculated the correlation against ECS. This has been elaborated in the revised version.

(Lines 131-133) "When averaging the seasonal dLCF/dTS over the oceanic regions between latitudinal belts of 30-60°N/S in each model, a high correlation of -0.82 between ECS and the seasonal dLCF/dTS is found across the 26 models (Fig. 2b)."

(Lines 395-396) "Seasonal dLCF/dTS over the extratropics in each model as shown in Fig. 2b is obtained by averaging the dLCF/dTS over all ocean grid points between 30-60°N/S."

9. L.132: You could calculate the correlation coefficient of these two correlation patterns to get a more quantitative statement than "greatly resembles".

Following this reviewer's suggestion, we calculated the pattern correlation, which is about 0.6. Considering the very large degree of freedom (spatial grid points), this is highly significant.

10. L.134-137: You should mention here that you are referring to a correlation across CMIP models and the "range" is an inter-model range.

We modified this sentence to (Lines 141-143) "although a more rapid change in LCF with surface temperature is found on the seasonal time scale compared to that associated with the long-term trend."

11. L.155-156: The sentence "[...] indicating that the high climate sensitivity with the ECS greater than 5K in several GCMs may be overestimated" seems to contradict your abstract which states that you find a "strong extratropical low cloud feedback that supports the high ECS models".

Thanks for these nice comments. To make these statements more accurate, we have made the following changes in the revised paper:

(Lines 156-158) "...which is closer to the high ECS models, although the seasonal dLCF/dTS tends to be largely overestimated in several extremely high ECS models."

For the statement in the abstract of original manuscript, we intended to mention that "We present further evidence of a strong extratropical low-cloud feedback using satellite observations of cloud radiative fluxes during the last decades that does not support the low ECS models." However, due to the word limit for the abstract by Nature Communications, we have decided to delete this sentence and add a discussion on the different cloud regimes in climate models as this reviewer suggested.

12. L.191-198: This entire paragraph about the short-term trend in the vertical velocity seems to

be out of context as this is only fully discussed in detail in the subsequent section.

The discussion on the short-term trend in the vertical velocity has been moved to the discussions on the physical mechanisms.

(Lines 266-279) “Considering a strong coupling between the lower-to-mid-tropospheric ascending motion and the vertically-extended clouds, a significant decreasing trend in the extratropical LCF during 1980-2014 in NCAR-CESM2 and HadGEM3-LL and other high ECS models (Fig. 4a) could be closely linked to the decreasing trend in the mid-tropospheric ascending motion as shown in Fig. 4c. While a decreasing trend in the extratropical ascending motion with recent warming is also evident in the low ECS models including MPI-ESM2-LR (Fig. 4c), the change in tropospheric vertical motion tends not to effectively affect low clouds in these models due to a decoupling between LCF and the vertically-extended clouds as previously discussed. As a result, the year-to-year variability of extratropical low clouds in the low ECS models such as MPI-ESM2-LR is largely controlled by the low-top cloud regime as in its seasonal cycle, with the absence of a significant decreasing trend in extratropical LCF during 1980-2014 (Fig. 4a, 5f). Note that while a decreasing trend in the ascending motion over the extratropics is simulated in both the high and low ECS models, it is not evident in the ERA5 reanalysis (Fig. 4c), possibly due to the strong internal variability in the observations and/or possible large deficiencies involved with the vertical velocity field in the reanalysis dataset.”

13. L. 267-270: I can only see the "significantly enhanced low clouds" in the CEMS2 and HadGEM3-LL models, but not in the CloudSat/CALIPSO observations.

Although not as significant as in CESM2, the increased frequency of enhanced low clouds during southern winter (May-Oct) is still evident in the CloudSat/CALIPSO observations based on the original Fig. S4. For demonstration purposes, we slightly changed the location of the subdomain over the Southern Oceans to make it more obvious (see new supporting Fig. 6). Also, we modified the discussions in the revised version as follows:

(Lines 255-258) “Significantly enhanced LCF or more frequent occurrence of low clouds during the austral winter (May-October) is found to be closely linked to vertically-extended clouds in the mid- and upper-troposphere in CESM2 and HadGEM3-LL, as also largely evident in CloudSat/CALIPSO observations. ”

14. L.302: "Fig. 5d" should probably be "Fig. 5e", "Fig. S7b4" should probably be "Fig. S7c4".

This part has been modified as follows:

(Lines 274-277) “As a result, the year-to-year variability of extratropical low clouds in the low ECS models such as MPI-ESM2-LR is largely controlled by the low-top cloud regime as in its seasonal cycle, leading to the absence of a significant decreasing trend in extratropical LCF during 1980-2014 (Figs. 4a, 5f).”

15. L. 306-307: Figs. 4a,c suggest this should probably be a "pronounced decrease with

decreasing tropospheric ascending motion", not an "increase".

Thanks. This has been modified.

(Lines 282-284) "A pronounced reduction of LCF with decreasing tropospheric ascending motion is found in the high ECS models through a strong modulation by the vertically-extended clouds,"

16. L.312-324: A relevant paper that should be mentioned in this paragraph is Grise and Medeiros (2016).

This reference has been added as suggested.

(Lines 288-289): "Motivated by the above cloud regime analyses, following a similar approach used in Grise and Medeiros (2016)⁴⁶, distinct processes involved with the LCF variability...."

17. L.589-595: Please explain what the vertical line and shading represent in the caption of Fig. 2b.

Thanks for pointing out this. This was overlooked in the original manuscript, should have been mentioned in details. Now this has been added in Fig. 2 caption and also discussed in Methods:

(Lines 672-677) "The light grey shaded areas around the regression line in b) represent the standard prediction errors by the linear fit following Schlund et al. (2020). The vertical red line in b) corresponds to the seasonal dLCF/dTS derived from the climatological seasonal cycles of LCF and TS based on the satellite observations for the period of 2006-2011, with its uncertainty (dark grey shading) estimated by the mean and one standard deviation of the observed seasonal dLCF/dTS in each year of 2006-2011."

(Lines 421-426) "While a seasonal dLCF/dTS of $-1.71\%/K$ over the extratropics from the CloudSat/CALIPSO observations is derived based on climatological seasonal cycle of vertical clouds, its year-to-year variability is relatively weak with a mean value of $-1.65\%/K$ and a standard deviation of $0.27\%/K$ during the six-year period. This mean value of the seasonal dLCF/dTS with plus/minus one standard deviation is used to estimate uncertainties involved with the observed seasonal dLCF/dTS (see the vertical grey bar in Fig. 2b)."

18. L.404-407: It would be helpful to describe this calculation step by step or provide a formula. Currently the order in which the calculations described here (averaging over the corresponding grid cells, averaging over the periods 2061-2095/1980-2014, calculating the difference, calculation the fraction) are carried out is not 100% clear to me. Also, why don't you use the linear trend for the long-term change similarly to the seasonal and short-term time scale?

This part has been modified. We believe descriptions of the method have been much improved.

(Lines 401-405) “To calculate the dLCF/dTS associated with the long-term trend over the extra-tropics in each model as shown in Fig. 2c,d, long-term changes in LCF (dLCF) and surface temperature (dTS) on each grid cell are first derived by the differences in their 35-year mean values between 2061-2095 from simulations under the shared socioeconomic pathway 5-8.5 (SSP585) scenario and 1980-2014 from historical simulations. Then dLCF and dTS are further averaged over all ocean grid points between 30-60°N/S, and the ratio of their spatially averaged values defined as the dLCF/dTS associated with the long-term trend.”

This approach will avoid very large values in dLCF/dTS over locations where dTS is very small if the dLCF/dTS is calculated over each grid point.

Similar regression approach can also be applied for derivation of dLCF/dTS associated with long-term trend. However, since the calculation of LCF needs 3D cloud profiles, to apply this approach we will need to download all the 3D cloud data for the entire 21st century for each model, which will involve much larger amount of data for downloading and processing. In this study, we decided to apply a slightly easy way to calculate dLCF/dTS associated with long-term trend.

19. Fig. S5: Your contour lines show negative values for the cloud fraction. Can you explain why this is the case? Are you showing differences to here?

These are anomalies relative to mean cloud profiles over each sub-domain over the Southern Oceans, since long-term climatology has been removed before the EOF.

(Lines 444-446) “An EOF analysis is then conducted based on the covariance matrix of a concatenated daily series of vertical cloud fraction anomalies over the 12 sub-regions (after removal of their corresponding long-term climatology over each sub-region) during the 35-year period ...”

Technical Corrections

1. The usage of the symbol "dTS" in the denominator of dLCF/dts might be more appropriate than "dts".

Modified as suggested. Thanks.

2. The term "low cloud variability" might be a bit misleading. Instead of variability of low clouds, one could also think of low variability in clouds. Thus, I would recommend using a different expression for that.

Thanks for pointing out this. We have made changes in the revised version.

3. L.112,394: I think a "cell" is missing after "grid".

Added “cell” as suggested.

4. L.589: LTS -> LCF.

Modified.

5. Fig. 4: Elements in the legends are missing in Panels (a)–(e) (e.g., the colored lines are missing).

These have been added in the revisions.

References

Most of the relevant papers have been added in the revised manuscript.

We again greatly appreciate this reviewer’s significant efforts and time in providing very comprehensive and insightful comments on the previous version of our manuscript. This leads to significant improvement of both the science and presentation of this study.

Responses to Reviewer #2:

Jiang et al. present an argument for a new emergent constraint that relates the spread in climate-model projections of equilibrium climate sensitivity (ECS) to the spread in climate-model estimates of the seasonal cycle of extratropical low-level clouds. The research topic is relevant to a broad scientific audience, and the writing and figures are clear. However, I believe that the paper lacks a convincing physical explanation for the emergent constraint, and it does not sufficiently discuss how the results relate to the existing literature. The method of uncertainty quantification and choice of observational data for the emergent constraint are also not sufficiently explained and justified. If these issues are addressed, then I believe that the paper may be suitable for publication. I recommend major revision.

We thank this reviewer for his/her insightful comments. The following is our point-by-point responses.

General Comments

- One of the most important shortcomings of the paper is the lack of a convincing physical explanation for the emergent constraint. The authors state that the seasonal cycle of extratropical storm-track clouds is linked to ECS, but they offer no specific physical argument for why this should be the case. Why does the seasonal variation of storm-track clouds provide predictive skill for how these clouds will change in a warming world? I would expect CO₂-driven warming and seasonal changes in insolation to cause different equator-to-pole temperature gradients and static stability changes, and therefore different variations in baroclinicity and storm-track activity. I think the authors need to explain this better. Furthermore, the authors base their arguments on relationships between clouds and local surface temperature, but the storm track is more sensitive to changes in equator-to-pole temperature gradients than local surface temperature. Thus, there seems to be an inconsistency between the interpretation that the authors provide and the relationships they examine. Please explain this apparent inconsistency.

We thank this reviewer for raising these concerns. We felt that there could be some misunderstanding. First of all, our results suggest that the seasonal cycle of extratropical LCF is highly linked to ECS, instead of *“the seasonal cycle of extratropical storm-track clouds is linked to ECS”* as this reviewer mentioned. In this study, we did not specifically emphasize the differences in the storm-track activity between the high and low ECS models. Instead, one of our main findings is that the extratropical LCF shows very different relationship to storm-track clouds (or vertically-extended clouds) between the high and low ECS models. In other words, as we shown in Supplementary Fig. 6, and also indicated by Supplementary Fig. 2e and Supplementary Fig. 3, on the seasonal time scale, the storm-track activity and associated deep clouds in the low ECS models are also enhanced during the winter season, which is largely similar to the high ECS models; however, due to a lack of coupling between LCF and the vertically-extended cloud modes, LCF in the low ECS models shows a minimum during the winter, instead of a winter maximum in the high ECS models in which LCF is highly coupled to storm track variability.

Therefore, the following questions, i.e., *“Why does the seasonal variation of storm-track clouds provide predictive skill for how these clouds will change in a warming world? I would expect*

CO2-driven warming and seasonal changes in insolation to cause different equator-to-pole temperature gradients and static stability changes, and therefore different variations in baroclinicity and storm-track activity.”, are also not directly relevant to the discussions in this study. Again, we mainly focused on the extratropical LCF in this study, not the dynamical processes over the extratropics, such as storm-track activity, jet stream shift, etc. As shown in Fig. 4c, associated with warming, the short-term trend in the vertical velocity over the extratropical regions suggests a decreasing trend in ascending motion in both high and low ECS models, despite their distinct trends in LCF (Fig. 4a). This can also be related to another question below from this reviewer on the insignificant correlation between ECS and dynamical processes reported in previous studies. Differences in the dynamical processes over the extratropics under a warming climate between the high and low ECS models, as this reviewer pointed out, however, are very interesting and warrant further investigations, but beyond the discussions of the main findings from this study.

Using local SST when deriving dLCF/dTS in this study is mainly to depict the seasonal cycle of extratropical LCF, or local cloud feedback associated with climate trend. As mentioned above, it is not the focus of this study on the future changes of storm-track variability or their differences in the high and low ECS models. Understanding changes in storm track activity under the warming climate could also be complicated, a topic itself worth a dedicated study in the future, since it involves with a competition between the surface and upper-tropospheric baroclinicity, and a tug of war in the warming trends between the tropics and the polar region (e.g., Shaw et al. 2016).

In this revised version, we have made significant changes to improve our presentation, particular on the underlying processes for the distinct extratropical low cloud feedback between the low and high ECS models. We hope this revised version well addresses this reviewer’s concern.

- One key lesson we learned from CMIP6 is that many of the emergent constraints developed from CMIP5 models do not hold up to scrutiny when applied to the CMIP6 ensemble. Now that we know this, I think it is important that any new emergent constraints are tested across both CMIP5 and CMIP6 models. I recommend that the authors examine if their emergent constraint holds true in the CMIP5 ensemble. If so, this would make the argument more convincing.

Thanks for this great suggestion. Following this suggestion, we also examined the relationship between the ECS and seasonal dLCF/dTS over the extratropics in 21 CMIP5 models (supporting Table S2), and showed this result in the Supplementary Fig. 7 along with the CMIP6 models. In general, results are consistent with the emergent constraint derived based on CMIP6 models, i.e., the relatively weak ECS in CMIP5 models is largely consistent with weak seasonal dLCF/dTS over the extratropics in CMIP5 models. This result confirms that the high ECS predicted in CMIP6 models is closely associated with changes in representation of extratropical low-cloud feedbacks in these models. Some detailed discussions on these results are added in the revised manuscript.

(Lines 336-347) “Since the emergent relationship between the seasonal dLCF/dTS over the extratropics and ECS in this study was derived based on CMIP6 models, it is interesting to verify whether this metric is also applicable for CMIP5 models. By plotting the seasonal dLCF/dTS and their corresponding ECS from 21 CMIP5 models (supporting Table S2) along with CMIP6 models (supporting Fig. S7), it is found that a majority of the CMIP5 models exhibit a much weaker seasonal cycle of extratropical LCF than the observations and the high ECS models from CMIP6, consistent with generally low ECS in the CMIP5 models. While a statistically significant correlation between ECS and the seasonal dLCF/dTS over the extratropics can still be obtained based on 26 CMIP6 and 21 CMIP5 models ($r \sim -0.56$, $p < 0.0001$), no significant correlation is found across the CMIP5 models alone ($r \sim 0.06$). This result suggests that the high ECS predicted in several CMIP6 models, which is responsible for the increase in the spread of ECS from CMIP5 to CMIP6, is closely associated with changes in representation of extratropical low-cloud feedbacks in these models, consistent with findings from other studies (Zelinka et al. 2020·Bacmeister et al. 2020·Bodas-Salcedo et al. 2019). “

- The paper would be stronger if the authors discuss how their results compare with the existing literature in more detail. For example, Grise and Polvani (2016) found weak or insignificant relationships between ECS and poleward shifts of the extratropical jet streams in a warming world across GCMs. Ceppi and Hartmann (2015) found a weak relationship between natural variations in latitude of the Southern Hemisphere extratropical jet and SW cloud radiative effects in observations. These findings seem to contradict the proposed emergent constraint. Please explain these apparent discrepancies.

This is also a very nice comment. Indeed, we feel it a weakness of this study that we could not provide a thorough understanding of the detailed processes underlying the reduction of extratropical LCF under a warming climate. While we found the reduction of extratropical low clouds under the future warming in high ECS models is associated with weakening of the ascending motion, it remains unknown about the relative roles of the Hadley Cell expansion, the mid-latitude storm-track variability, and the shift of the westerly Jet Stream. Regarding several previous studies on the insignificant relationship between model ECS or SWCRE with Jet shifts as this reviewer mentioned, we think these results do not necessarily contradict the findings in this study. Discussions on some possible reasons for these “discrepancies” have also been added in the revised manuscript (Lines 348-367, also copied below).

(Lines 348-367) “The cloud regime analysis suggests that the rapid reduction of extratropical low clouds under a warming climate in the high ECS models could be mainly due to the reduction of vertically-extended clouds in association with the weakening of ascending motion over the extratropics. However, future investigations are needed to better understand how the reduced vertical clouds and associated weakening of the ascending motion over the extratropics under a warming climate are linked to changes of the Hadley Cell expansion, the mid-latitude storm-track variability, and the shift of the westerly Jet Stream. A weak relationship between extratropical SWCRE or ECS and poleward shifts of the extratropical jet streams has been previously reported in observations and CMIP5 models^{45,52}. On one hand, this could be due to a significant role of the Hadley Cell expansion on extratropical LCF in addition

to the shift of the Jet Stream⁵². Meanwhile, this can also be ascribed to different model responses of extratropical low clouds to changes in environmental conditions as suggested by this study. For example, a weakening of the extratropical ascending motion is simulated in both the high and low ECS models (Fig. 4c), whereas the reduced extratropical LCF is only simulated in the high ECS models (Fig. 4a). Since CMIP5 models exhibit large deficiencies in representing the seasonal evolution of extratropical low clouds in general, investigations based on the high ECS models from CMIP6 are expected to provide important insights on how extratropical LCF and its induced radiative effects respond to changes of the dynamical processes over the extratropics. Also note that considering large model deficiencies in representing extratropical storm-track clouds in the low ECS models as shown in this study, it needs to be cautious when assessing environmental changes over the extratropics under a future climate based on these low ECS models, including most of the CMIP5 models.”

- The uncertainty quantification needs to be described. In particular, how is spatial autocorrelation between the grid boxes accounted for? If you assume that every grid box is independent, then you will underestimate uncertainty. I recommend accounting for spatial autocorrelation following a method similar to that of Myers et al. (2021). Also, the authors show that their observational quantity that determines the emergent constraint ($dLCF/dts$) depends on the choice of observational dataset used. However, the authors only show one of the observational datasets in the vertical line in Fig. 2b. Why not show all of the results? Without showing all results or explaining why you picked just one, you risk sounding like you are selectively choosing evidence that produces the highest ECS, which would be problematic. Please justify your choices more clearly in the text.

In Myers et al. (2021), the near-global low cloud feedback is constrained using satellite observations of low cloud changes with various cloud controlling factors on each grid point. In this case, the dependence among different grid boxes (effective number of grid points) is accounted when estimating uncertainty using an auto-correlation approach. In this study, the seasonal $dLCF/dTS$ over the extratropics in each model and observations is derived by an averaged value of $dLCF/dTS$ over 30-60°N/S, and regressed onto the ECS across 26 models, instead of regression over each grid points of the extratropics. Therefore, the approach in this study is different from Myers et al. (2021), and we simply estimated the observational uncertainty of the seasonal $dLCF/dTS$ based on the year-to-year variability of $dLCF/dTS$. Based on CloudSat/CALIPSO observations during the period of 2006-2011, the mean value of the seasonal $dLCF/dTS$ is -1.65%/K with a standard deviation of 0.27%/K during the six-year period, while the seasonal $dLCF/dTS$ based on the climatological seasonal cycle derived from the entire 6-year period is -1.71%/K. These information on observational uncertainty estimation has been added in the revised manuscript in both “Methods” and caption for Fig. 2.

(Lines 421-429) “While a seasonal $dLCF/dTS$ of -1.71%/K over the extratropics from the CloudSat/CALIPSO observations is derived based on climatological seasonal cycle of vertical clouds, its year-to-year variability is relatively weak with a mean value of -1.65%/K and a standard deviation of 0.27%/K during the six-year period. This mean value of the seasonal $dLCF/dTS$ with plus/minus one standard deviation is used to estimate uncertainties involved

with the observed seasonal dLCF/dTS (see the vertical grey bar in Fig. 3a). Sensitivity tests based on model simulations suggest that the climatological seasonal cycle of vertical clouds and thus the seasonal dLCF/dTS derived based on model simulations during the six-year period of 2006-2011 are largely identical to that derived from the entire 35-year period of historical simulations as shown in Fig. 2. This lends confidence in constraining the simulated seasonal dLCF/dTS over the extratropics using the CloudSat/CLIPSO observations.”

(Lines 672-677) “The light grey shaded areas around the regression line in b) represent the standard prediction errors by the linear fit following Schlund et al. 2020. The vertical red line in b) corresponds to the seasonal dLCF/dTS derived from the climatological seasonal cycles of LCF and TS based on the satellite observations for the period of 2006-2011, with its uncertainty (dark grey shading) estimated by the mean and one standard deviation of the observed seasonal dLCF/dTS in each year of 2006-2011.”

Regarding the selection of the observational dataset to derive the seasonal dLCF/dTS, this is also related to the concerns by Reviewer 1 – we largely copied our previous responses below. It has been well known that the CloudSat/CALIPSO combined cloud product (2B-GEOPROFLIDAR) is the best dataset in depicting the clouds associated with the extratropical cyclones (Naud et al. 2010; Govekar et al. 2011; Li et al. 2014; Mace and Zhang 2014). As shown in our original manuscript, even the active LIDAR observations of vertical clouds by CALIPSO fail in detecting the vertically-extended extratropical clouds (see Fig. A1), other passive sensors such as ISCCP will have more serious deficiencies in depicting the low clouds associated with the vertically-extended storm-track clouds due to their limitations in detecting low clouds collocated with deep or multi-layered clouds, as illustrated in (Bodas-Salcedo et al. 2014). The passive sensing also has difficulties in accurately detecting the cloud-top height (Bodas-Salcedo et al. 2014). Therefore, there is no doubt the combined CloudSat/CALIPSO dataset provides the best observations to derive the dLCF/dTS in this study.

Considering that a paper to be published at Nature Communications will be mainly targeted at general readers who not necessarily need to know such detailed information regarding the uncertainties involved with different satellite sensors, and to concentrate this paper mainly on scientific discussions, in this revised manuscript, we decided to remove discussions on observational uncertainties in the main text. Instead, we provide a brief introduction of the combined CloudSat/CALIPSO cloud observations in “Methods”, why it is the best dataset available for vertical cloud observations over other datasets, particularly for studies on clouds associated with extratropical cyclones.

Corresponding modifications on this part include:

(Lines 413-418) “The observational dataset for vertical cloud profiles is based on a combined product from the CloudSat and Cloud-Aerosol Lidar and Infrared Pathfinder Satellite Observation (CALIPSO) satellites (2B-GEOPROFLIDAR; version P2R05)²⁸. The combined CloudSat/CALIPSO dataset has been considered the best satellite observations for vertical cloud structures associated with extratropical cyclones^{55,36,56,43}, mainly due to its advantage of the

Cloud Profiling Radar (CPR) aboard CloudSat in detecting optically thick hydrometeor layers, and the CALIPSO lidar in detecting optically thin cloud layers that could be missed by the CPR^{57,58,28}. “

(Lines 430-436) “It is also noteworthy that global low-cloud observations are also provided by several passive-sensing satellites, such as the International Satellite Cloud Climatology Project (Rossow et al. 2022). While these passive instruments provide useful information of horizontal distribution of low-top clouds, they have limitations in detecting low clouds collocated with deep or multi-layered clouds, such as those associated with extratropical cyclones along the mid-latitude storm tracks. The passive sensing also has difficulties in accurately detecting the cloud-top height (Bodas-Salcedo et al. 2014). Therefore, the low-cloud fractions based on these passive-sensing satellite observations are not supposedly to be directly compared to those derived from the CloudSat/CALIPSO observations and model simulations in this study.”

Specific Comments

- Line 76: “generate” -> “generates”

Modified.

- Line 77: “to amplify” -> “that amplifies”

Modified as suggested.

- Line 155: please state the confidence level for the ECS uncertainty range (one sigma, 90% CI, 95% CI, etc.)

As suggested by the Reviewer 1, we decided not to explicitly provide a range for the ECS in this revised version. This sentence has been removed in the revised manuscript.

- Line 189: the first CERES measurements are from 1997 from the TRMM satellite. The model trends are computed starting in 1980, which is much earlier. It would be good to revise this sentence so it doesn't sound like you are overstating the degree of agreement between the CERES trends and the model trends.

We made modifications of this sentence trying to make the statement not too strong.

(Lines 190-192) “While an increasing trend in SWCRE during 1980-2014 is found in the high ECS models, which tends to be supported by the Clouds and the Earth's Radiant Energy System (CERES) satellite observations, no significant trend is detected in the low ECS models (Fig. 4d).”

- Fig 4 (a-d): This figure would be clearer if you plot all of the values in each panel on the same y-axis. (i.e. all of the LCF data in Fig. 4a are plotted with the same y axis, all of the TS data in

Fig. 4b are plotted with the same y axis, etc.)

Thanks for this good suggestion. We modified panels a)-d) in this figure by showing anomalies of each variable after extracting their corresponding long-term climatology, so the short-term trends between the high and low ECS models can be directly compared.

- Line 196: The same criticisms you make about biases in the vertical velocity data from reanalysis apply to the vertical velocity data from climate models. I recommend rephrasing this sentence so that it doesn't imply that biases in reanalysis vertical velocity are worse than biases in global climate models, which is not true.

We agree with this reviewer that this statement needs to be clarified. What we mainly wanted to mention was that while the vertical velocity in the ERA5 reanalysis is regarded as "observations" here, it could just largely follow ECMWF-IFS model physics due to a lack of direct observational constraint for the vertical velocity. In the revised manuscript, we made the following modifications (Lines 277-279). Thanks for the very good point.

(Lines 277-279) "Note that while a decreasing trend in the ascending motion over the extratropics is simulated in both the high and low ECS models, it is not evident in the ERA5 reanalysis (Fig. 4c), possibly due to a lack of observational constraints involved with the vertical velocity field in the reanalysis dataset. "

- Line 218: It would be clearer if you could specifically state the large-scale cloud-controlling factors that are similar between the high-ECS and low-ECS models so that the reader doesn't have to check the supporting information.

This is another very good point. We have made modification following this reviewer's suggestion.

(Lines 210-214) "Despite the distinct seasonal variations in extratropical LCF between the high and low ECS models, several large-scale cloud controlling factors that have been widely used to understand low-cloud variations^{12,14}, including surface temperature, the estimated inversion strength (EIS), lower-tropospheric relative humidity and vertical velocity, exhibit largely similar seasonal evolution features between the two model groups (Supplementary Fig. 2)."

- Fig. 5: It would be clearer to plot (a), (c), and (e) on the same y-axis scale, or least use the same range of values in the y axis of each panel (e.g. y axis in (a) ranging from 45 to 70, y axis in (c) ranging from 25 to 50, etc.). Also, please explain why you chose to plot EOF mode 2 in the top row, EOF mode 2 and 3 in the middle row, and EOF mode 3 in the bottom row. This seems unnecessarily complicated and makes it difficult to compare across the rows.

Following this reviewer's suggestion, in the revised manuscript we also just plotted anomalous LCF in Fig. 5 by excluding their corresponding long-term climatology, so that amplitudes

between different models can be directly compared. The reason for showing the reconstructed LCF based on EOF2 for CESM2, EOF2+3 for HadGEM3-LL, and EOF3 for MPI-ESM2-LR is to illustrate different dominant modes in controlling the LCF variability in these modes. Along with the Supplementary Fig. 5, the Fig. 5 in the main text serves as a show-case to demonstrate that the LCF variability tends to be coupled with vertically-extended clouds in the high ECS models, while this coupling is absent in the low ECS models, which largely motivates the composite analysis in Fig. 6.

We thank this reviewer again for his/her critical comments particularly regarding the relationship between extratropical LCF and dynamical processes, which motivated some important discussions and led to much improvement of this revised manuscript. How dynamical processes regulate the extratropical LCF variability and why the high and low ECS models respond in a different way indeed will be a very interesting topic for our future study.

REVIEWERS' COMMENTS

Reviewer #1 (Remarks to the Author):

Review on „Muted extratropical low cloud seasonal cycle is closely linked to lower model climate sensitivity“

I very much appreciate the efforts that the authors have undertaken to address my comments raised in the previous round of reviews. In particular, the more detailed description of their methods and enhancements to the figures improved the comprehensibility and readability of the paper a lot. I only have some minor comments below that could be used to further improve the manuscript. Nevertheless, I fully recommend this study for publication in Nature Communications.

Well done, congratulations to the paper!

Minor comments

1. I am not sure why you decided to change the title of the manuscript. For me, one of the main conclusions of the paper is that low-ECS models might underestimate ECS (as stated in your result section in l.160 and your conclusions in l.320). Your current title just says “weak extratropical low cloud seasonal cycle -> low ECS”; there is no mention that observations suggest a stronger extratropical low cloud seasonal cycle, and thus are more consistent with the high-ECS models. In my opinion, since you dedicate a large part of the paper to those observations, this possible underestimation of ECS should be mentioned in the title (maybe just use the old version of the manuscript?). It would be also good to explicitly mention this underestimation in the abstract with 2-3 words if the abstract word limit allows that.
2. Thank you for updating the methods section “Cloud regime analysis”, it is much clearer now. Would it be possible to slightly elaborate on the sentence “Moreover, 3D structure in cloud and vertical velocity anomalies associated with each leading mode can also be derived based on lag-0 regressions against their corresponding daily PCs” (l.455,456)? I am not sure I fully understand that in the current form. For example, do Figures S4b-d (and others) show only 12 data points on the longitude axis (for each sub-region)? The figure looks like a finer resolution is used.

Specific Comments

1. L.242-243: Which figure shows that “the combined mid- and low-top cloud regimes [...] are in-phase with each other in HadGEM3-LL”?
2. L.423: Are you really referring to the “mean” and “standard deviation” of the “year-to-year variability” here or is it actually the mean and standard deviation of 6 different 1-year climatologies? Only the latter (standard deviation) sounds like a variability to me, so it might be helpful to rephrase this here.

Technical Corrections

1. L.44, 59, caption of Fig. S1: “surface” -> “near-surface”.
2. L.377: “contributed” -> e.g., “determined”.
3. L.443: “is” -> “are”.
4. L.770: “year-to-year variability” -> “time series”.
5. L.771: “departures” -> e.g., “deviations”.

Reviewer #2 (Remarks to the Author):

I would like to thank the authors for considering my comments and revising the manuscript accordingly. My initial concerns have been adequately addressed. I recommend that the manuscript be accepted for publication.

Again we thank the two reviewers for their great efforts and very insightful comments that significantly improved this manuscript. The following are our point-by-point responses to the latest comments. The review comments are shown in black with our responses in blue.

Responses to Reviewer #1:

1. I am not sure why you decided to change the title of the manuscript. For me, one of the main conclusions of the paper is that low-ECS models might underestimate ECS (as stated in your result section in l.160 and your conclusions in l.320). Your current title just says “weak extratropical low cloud seasonal cycle -> low ECS”; there is no mention that observations suggest a stronger extratropical low cloud seasonal cycle, and thus are more consistent with the high-ECS models. In my opinion, since you dedicate a large part of the paper to those observations, this possible underestimation of ECS should be mentioned in the title (maybe just use the old version of the manuscript?). It would be also good to explicitly mention this underestimation in the abstract with 2-3 words if the abstract word limit allows that.

As suggested by this reviewer, we further modified the title of the manuscript to “Muted extratropical low cloud seasonal cycle is closely linked to underestimated climate sensitivity in models”.

In the abstract, however, we could not add one more sentence to explicitly mention the potential underestimate of ECS in the low ECS models due to the limit of words. We think it will be fine since this has been implied by the following sentence in the abstract and particularly it has been explicitly mentioned in the title.

“In contrast, a pronounced seasonal cycle of extratropical LCF, as supported by satellite observations, is largely absent in models with $ECS < 3.3K$.”

2. Thank you for updating the methods section “Cloud regime analysis”, it is much clearer now. Would it be possible to slightly elaborate on the sentence “Moreover, 3D structure in cloud and vertical velocity anomalies associated with each leading mode can also be derived based on lag-0 regressions against their corresponding daily PCs” (l.455,456)? I am not sure I fully understand that in the current form. For example, do Figures S4b-d (and others) show only 12 data points on the longitude axis (for each sub-region)? The figure looks like a finer resolution is used.

We thank this reviewer for this nice comment. The following is the modified version (please note fine resolution of 3D structures can be obtained using fine resolution of cloud and vertical velocity when calculating the regressions).

“Detailed 3D structures in cloud and vertical velocity anomalies associated with each leading mode can also be derived based on lag-0 regressions of these daily anomalous fields (2.5x2.5deg, 19 pressure levels, similarly with their corresponding long-term climatology on each grid removed) against the corresponding daily PCs over each sub-region, and then by compositing these structures over the 12 sub-regions with respect to the center of each sub-region, e.g. longitude 0 in Supplementary Fig. 4. “

Specific Comments

1. L.242-243: Which figure shows that “the combined mid- and low-top cloud regimes [...] are in-phase with each other in HadGEM3-LL”?

The in-phase relationship between the mid- and low-top cloud regimes in HadGEM3-LL can be seen from Supplementary Fig. 5i,5j. We added this information in the revised manuscript.

2. L.423: Are you really referring to the “mean” and “standard deviation” of the “year-to-year variability” here or is it actually the mean and standard deviation of 6 different 1-year climatologies? Only the latter (standard deviation) sounds like a variability to me, so it might be helpful to rephrase this here.

Indeed, the latter is what we meant. In the revised version, we made modifications as follows:

“While a seasonal dLCF/dTS of $-1.71\%/K$ over the extratropics from the CloudSat/CALIPSO observations is derived based on climatological seasonal cycle of vertical clouds (Fig. 2b, vertical red line), its corresponding value derived from each individual year during the six-year period shows relatively weak year-to-year variability with a mean value of $-1.65\%/K$ and a standard deviation of $0.27\%/K$.”

Technical Corrections

1. L.44, 59, caption of Fig. S1: “surface” -> “near-surface”.

Modified as suggested.

2. L.377: “contributed” -> e.g., “determined”.

Modified as suggested.

3. L.443: “is” -> “are”.

Modified as suggested.

4. L.770: “year-to-year variability” -> “time series”.

Modified as suggested.

5. L.771: “departures” -> e.g., “deviations”.

Modified as suggested.